# AN INFORMATION-THEORETIC ANALYSIS OF THOMPSON SAMPLING FOR LOGISTIC BANDITS

## ABSTRACT

We study the performance of the Thompson Sampling algorithm for logistic bandit problems, where the agent receives binary rewards with probabilities determined by a logistic function, $\exp(\beta\langle a, \theta\rangle)/(1 + \exp(\beta\langle a, \theta\rangle))$, with slope parameter $\beta$. We focus on the setting where both the action $a$ and parameter $\theta$ lie within the $d$-dimensional unit ball. Adopting the information-theoretic framework introduced by (Russo & Van Roy, 2015), we analyze the information ratio, a statistic that quantifies the trade-off between the information gained about the optimal action and the immediate regret incurred. We improve upon previous results by establishing that the information ratio is bounded by $\frac{9}{2}d\alpha^{-2}$, where $\alpha$ is a minimax measure of the alignment between the action space and the parameter space, independent of $\beta$. Notably, we derive a regret bound of order $O(d/\alpha\sqrt{T\log(\beta T/d)})$, which scales only logarithmically with the logistic function parameter $\beta$. To the best of our knowledge, this is the first regret bound for logistic bandits that achieves logarithmic dependence on $\beta$ while being independent of the number of actions. In particular, when the action space encompasses the parameter space, the expected regret of Thompson Sampling is of order $\tilde{O}(d\sqrt{T})$.

## 1 INTRODUCTION

This paper studies the logistic bandit problem, where for $T$ time steps, an agent selects an action and receives binary rewards with probabilities determined by the logistic function $\exp(\beta\langle a, \theta\rangle)/(1 + \exp(\beta\langle a, \theta\rangle))$ with slope parameter $\beta > 0$. In this setting, both the action $a \in \mathcal{A}$ and the parameter vector $\theta \in \mathcal{O}$ lie within the $d$-dimensional unit ball. Logistic bandits are used to model various scenarios, such as click-through rate prediction, spam email detection, and personalized advertisement systems, where, in the latter case, content is suggested to users who provide binary feedback, such as "like" or "dislike" (Chapelle & Li, 2011; McMahan & Streeter, 2012; Russo & Van Roy, 2017).

The performance, or regret, of algorithms for logistic bandits has been extensively studied, with significant contributions including analyses of Upper-Confidence-Bound (UCB) algorithms by (Filippi et al., 2010), (Li et al., 2017), and (Faury et al., 2020) as well as the study of Thompson Sampling (TS) by (Russo & Van Roy, 2014a) and (Abeille & Lazaric, 2017). However, nearly all existing regret bounds exhibit an exponential dependence on the parameter $\beta$ (see our comparison in Table 1). This dependence is unsatisfactory because, in practice, as $\beta$ increases, it can be faster to identify the optimal action as the distinction between near-optimal and sub-optimal actions is more pronounced.

In this work, we focus on the Thompson Sampling algorithm (Thompson, 1933), which, despite its simplicity, has proven to be highly effective across a wide range of problems (Russo et al., 2018; Chapelle & Li, 2011). To analyze the regret of Thompson Sampling, (Russo & Van Roy, 2015) introduced the concept of the information ratio, defined as the ratio between the squared expected difference between the optimal and actual rewards, and the information gained about the optimal action. (Dong & Van Roy, 2018) conjectured that the Thompson Sampling information ratio for logistic bandits could be bounded solely by the problem's dimension $d$, and several studies have since aimed to characterize this ratio.

Recently, (Neu et al., 2022) derived a regret bound of the order $O(\sqrt{dT|\mathcal{A}|\log(\beta T)})$ on the logistic bandit problem, but their result relies on a worst-case information ratio bound scaling with the cardinality of the action space $|\mathcal{A}|$ and their regret bound becomes vacuous for problems with

| Algorithm | Regret Upper Bound | Note |
|:---:|:---:|:---:|
| Thompson Sampling (Russo & Van Roy, 2014a) | $O\left(e^{\beta} \cdot d \cdot T^{1/2} \cdot \log(T)^{3/2}\right)$ | Bayesian bound |
| Thompson Sampling (Abeille & Lazaric, 2017) | $O\left(e^{\beta} \cdot d^{3/2} \cdot \log(d)^{1/2} \cdot T^{1/2} \log(T)^{3/2}\right)$ | Frequentist bound |
| Thompson Sampling (Dong & Van Roy, 2018) | $O\left(e^{\beta} \cdot d \cdot T^{1/2} \log(T/d)^{1/2}\right)$ | Bayesian bound |
| Logistic-UCB-2 (Faury et al., 2020) | $O\left(d \cdot T^{1/2} \cdot \log(T) + e^{\beta} \cdot d^2 \cdot \log(T)^2\right)$ | Frequentist bound |
| Thompson Sampling (*this paper*) | $O\left(d/\alpha \cdot T^{1/2} \cdot \log(T\beta/d)^{1/2}\right)$ | Bayesian bound, $\alpha$ is independent of $\beta$ (defined in Section 2) |

Table 1: Comparison of various regret guarantees for the logistic bandit problem.

continuous or infinite action space even though Thompson Sampling is known to perform well under these settings (Russo & Van Roy, 2014a). Studying the Thompson Sampling regret for logistic bandits, (Dong et al., 2019) introduced two statistics to characterize the sets $\mathcal{A}$ and $\mathcal{O}$, the *minimax alignment constant*[1] $\alpha = \min_{\theta \in \mathcal{O}} \max_{a \in \mathcal{A}} \langle a, \theta \rangle$ and the *fragility dimension* $\eta$, which is the cardinality of the largest subset of parameters such that their corresponding optimal action is misaligned with any other parameter from the subset. Using those statistics, they showed that for $\beta < 2$, the information ratio for Thompson Sampling is bounded by $100 \max(d, \eta)\alpha^{-2}$. They also suggested, through numerical computations, that this bound holds for larger values of $\beta$. However, their work has two key limitations. First, they do not provide a rigorous proof for generalizing their bound to larger values of $\beta$. Second, and more critically, their regret analysis is incorrect as it relies on the rate-distortion bound from (Dong & Van Roy, 2018), which is incompatible with a bound on the Thompson Sampling information ratio. Indeed, the regret analysis in Dong & Van Roy (2018) specifically requires a bound on the one-step compressed Thompson Sampling information ratio, which is a fundamentally different quantity from the Thompson Sampling information ratio studied in the work of (Dong et al., 2019). We elaborate on these gaps in more detail in Appendix E.

In this work, we address these issues and obtain a regret bound that scales only logarithmically with the slope of the logistic function, while remaining independent of the cardinality of the action space. Our key contributions are as follows:

- We propose an information-theoretic regret bound of order $O(\sqrt{T\Gamma(\mathrm{H}(\Theta_{\varepsilon}) + \beta\varepsilon T)})$ that holds for infinite and continuous action and parameter spaces. The bound and relies on the entropy of the quantized parameter, $\mathrm{H}(\Theta_{\varepsilon})$, and the average expected information ratio of Thompson Sampling, $\Gamma$. This result is achieved by adapting the approaches from (Neu et al., 2021) and (Gouverneur et al., 2023) to the logistic bandit setting.

- We present a refined analysis showing that, for all $\beta > 0$, the information ratio of Thompson Sampling for logistic bandits is bounded by $\frac{9}{2}d\alpha^{-2}$, improving upon the previous results. Notably, our bound does not depend on the fragility dimension $\eta$.

- We establish a regret bound of order $O(d/\alpha\sqrt{T \log(\beta T/d)})$ for Thompson Sampling. To our knowledge, this is the first bound on logistic bandits that scales only logarithmically with $\beta > 0$ and is independent of the number of actions. Additionally, we show that if the action space encompasses the parameter space or if the action space is a design parameter, the expected TS regret is bounded in $O(d\sqrt{T \log(\beta T/d)})$ with no dependence on $\alpha$.

The rest of the paper is organized as follows. Section 2 introduces the logistic bandit problem setup, defines the Bayesian expected regret, and the specific notations used. Section 3 introduces the Thompson Sampling algorithm and the information ratio analysis. Section 4 states and discusses our main results, providing the improved regret bounds. Section 5 presents the key ideas of our analysis for studying the information ratio, and finally, Section 6 discusses our results and future extensions.

---

[1]This statistic is referred to as the *worst-case optimal log-odds* in the work of (Dong et al., 2019).

## 2 PROBLEM SETUP

We consider a logistic bandit problem, where at each time step $t \in \{1, \ldots, T\}$, an agent selects an action $A_t$ and receives a binary reward $R_t \in \{0, 1\}$ with probability following a logistic function:

$$\mathbb{P}(R_t = 1 | A_t = a, \Theta = \theta) = \frac{\exp(\beta\langle a, \theta\rangle)}{1 + \exp(\beta\langle a, \theta\rangle)},$$

where $\beta > 0$ is a known scale parameter, and $\langle a, \theta \rangle$ denotes the inner product of the action vector $a \in \mathcal{A}$ and the unknown parameter $\theta \in \mathcal{O}$. Throughout the paper, we denote the logistic function as $\phi_\beta(\langle a, \theta\rangle)$. The probability of obtaining a reward is maximized when the inner product between the action and environment parameters is maximized. In our setting, both the action $a$ and the parameter vector $\theta$ lie within the $d$-dimensional Euclidean unit ball, $\mathbf{B}_d(0, 1)$. Note that this is equivalent to the problem considered by (Faury et al., 2020) using $\beta$ as the maximal norm for the parameters $\theta \in \mathcal{O}$.

For a given action space $\mathcal{A}$ and parameter space $\mathcal{O}$, we define their *minimax alignment constant* $\alpha := \min_{\theta \in \mathcal{O}} \max_{a \in \mathcal{A}} \langle a, \theta\rangle$. For the rest of the paper, we assume that the action and parameters spaces are such that $\alpha \geq 0$. This assumption is relatively mild, as it suffices for the action space $\mathcal{A}$ to contain two opposed actions $a, a'$ (i.e. $a = -a'$) to ensure $\alpha \geq 0$ for any parameter set $\mathcal{O} \subseteq \mathbf{B}_d(0, 1)$.

Following the Bayesian framework, we assume the parameter vector $\theta \in \mathcal{O}$ is sampled from a known prior distribution $\mathbb{P}_\Theta$. This prior, together with the reward distribution $\mathbb{P}_{R|A,\Theta}$, fully describes the logistic bandit problem. As the reward distribution depends only on the selected action and the parameter, it can be expressed as $R_t = R(A_t, \Theta)$ for some possibly random function $R : \mathcal{A} \times \mathcal{O} \to \mathbb{R}$. The agent's history at time $t$ is denoted by $H^t = \{A_1, R_1, \ldots, A_{t-1}, R_{t-1}\}$, representing all past actions and rewards observed up to time $t$.

The goal of the agent is to sequentially select actions that maximize the total accumulated reward, or equivalently, that minimize the total expected regret defined as:

$$\mathbb{E}[\text{Regret}(T)] := \mathbb{E}\left[\sum_{t=1}^{T} R(A^\star, \Theta) - R(A_t, \Theta)\right],$$

where $A^\star$ is the *optimal action* for the corresponding parameter $\Theta$. We construct the optimal mapping $\pi_\star(\theta) := \arg\max_{a \in \mathcal{A}} \mathbb{E}[R(a, \theta)]$ and we can write $A^\star = \pi_\star(\Theta)$. To ensure such a mapping exists, we make the technical assumption that the set of actions $\mathcal{A}$ is compact. Following (Dong et al., 2019), we assume without loss of generality that the mapping $\pi_\star$ is one-to-one[2].

Since the $\sigma$-algebras of the history are often used in conditioning, we introduce the notations $\mathbb{E}_t[\cdot] := \mathbb{E}[\cdot | H^t]$ and $\mathbb{P}_t[\cdot] := \mathbb{P}[\cdot | H^t]$ to denote the conditional expectation and probability given the history $H^t$, respectively. Additionally, we define $\mathrm{I}_t(A^\star; R_t | A_t) := \mathbb{E}_t[\mathrm{D}_{\mathrm{KL}}(\mathbb{P}_{R_t|H^t,A^\star,A_t} \| \mathbb{P}_{R_t|H^t,A_t})]$ as the disintegrated conditional mutual information between the optimal action $A^\star$ and the reward $R_t$ conditioned on the action $A_t$, *given the history $H^t$*.

## 3 THOMPSON SAMPLING AND INFORMATION RATIO

An elegant algorithm for solving bandit problems is the *Thompson Sampling* algorithm. It works by randomly selecting actions according to their posterior probability of being optimal. More specifically, at each time step $t \in \{1, \ldots, T\}$, the agent samples a parameter estimate $\hat{\Theta}_t$ from the posterior distribution conditioned on the history $H^t$ and selects the action that is optimal for the sampled parameter estimate, $A_t = \pi_\star(\hat{\Theta}_t)$. The pseudocode for Thompson Sampling is given in Algorithm 1.

Studying the regret of Thompson Sampling in bandit problems, (Russo & Van Roy, 2015) introduced a key quantity to the analysis, the *information ratio* defined as the following random variable:

$$\Gamma_t := \frac{\mathbb{E}_t[R(A^\star, \Theta) - R(A_t, \Theta)]^2}{\mathrm{I}_t(A^\star; R(A_t, \Theta), A_t)}.$$

---

[2]Recall that Thompson Sampling disregards actions that are not optimal for any parameter. If a particular parameter is optimal for multiple actions, we can arbitrarily fix the mapping of that parameter to one of the optimal actions. Conversely, if a particular action is optimal for multiple parameters, we can introduce duplicate action labels to ensure a one-to-one correspondence between each parameter and its optimal action label. A rigorous explanation of this construction is provided in Appendix F.

---

**Algorithm 1** Thompson Sampling algorithm

---

1: **Input:** parameter prior $\mathbb{P}_\Theta$.
2: **for** $t = 1$ **to T do**
3:     Sample a parameter estimate $\hat{\Theta}_t \sim \mathbb{P}_{\Theta|H^t}$.
4:     Take the corresponding optimal action $A_t = \pi_\star(\hat{\Theta}_t)$.
5:     Collect the reward $R_t = R(A_t, \Theta)$.
6:     Update the history $H^{t+1} = H^t \cup \{A_t, R_t\}$.
7: **end for**

---

This ratio measures the trade-off between minimizing the current squared regret and gathering information about the optimal action; a small ratio indicates that a substantial gain of information about the optimal action compensates for any significant regret.

Russo and Van Roy use this concept to provide a general regret bound that depends on the time horizon $T$, the entropy of the prior distribution of $A^\star$, and an algorithm- and problem-dependent upper bound $\Gamma$ on the average expected information ratio (Russo & Van Roy, 2015, Proposition 1).

A limitation of this approach is that the prior entropy of the optimal action, $H(A^\star)$, can grow arbitrarily large with the number of actions or get infinite if the action space is continuous. We address this issue with Theorem 1, where we propose a regret bound that depends instead on the entropy of $\Theta_\varepsilon$, a quantized version of the parameter $\Theta$ while still being compatible with bounds on the Thompson Sampling information ratio.

## 4   MAIN RESULTS

This section presents our main results on the regret of Thompson Sampling for logistic bandits. In Theorem 1, we leverage the previously introduced concepts to derive an information-theoretic regret bound for logistic bandits that holds for continuous and infinite parameter spaces. Following this, we state in Proposition 1 our principal contribution, where we prove a bound of $\frac{9}{2}d\alpha^{-2}$ on the Thompson Sampling information ratio. By combining this result with our regret bound, we derive in Theorem 2, a bound on the expected regret of Thompson Sampling for logistic bandits, which scales as $O(d/\alpha\sqrt{T\log(\beta T/d)})$.

Our first theorem provides a regret bound that holds for large and continuous action spaces. Notably, the theorem uses bounds on the average expected information ratio of the *"standard"* Thompson Sampling, rather than the one-step compressed Thompson Sampling, as in (Dong & Van Roy, 2018, Theorem 1). This distinction is crucial, as the intricate construction of one-step compressed Thompson Sampling makes it challenging to derive bounds on its information ratio. The Thompson Sampling regret is related to the entropy of the quantized parameter $\Theta_\varepsilon$, which is defined as the closest approximation for $\Theta$ (as measured by the metric $\rho$) on an $\varepsilon$-net for $(\mathcal{O}, \rho)$.

**Definition 1** *Let the set $\mathcal{O}_\varepsilon$ be an $\varepsilon$-net for $(\mathcal{O}, \rho)$ with associated projection mapping $q : \mathcal{O} \to \mathcal{O}_\varepsilon$ such that for all $\theta \in \mathcal{O}$ we have $\rho(\theta, q(\theta)) \leq \varepsilon$. We define the* quantized parameter *as $\Theta_\varepsilon := q(\Theta)$.*

The proof of Theorem 1 adapts the techniques of of (Gouverneur et al., 2023, Theorem 2) and (Neu et al., 2021, Theorem 2) to the logistic bandits setting. It relies on a Lipschitz approximation of the conditional mutual information $I(\Theta; R_t|A_t, H^t)$ as $I(\Theta_\varepsilon; R_t|A_t, H^t) + \beta\varepsilon$ exploiting the fact that, for all $a \in \mathcal{A}$, the log-likelihood of $R(a, \theta)$ is $\beta$-Lipschitz with respect to $\theta \in \mathcal{O}$.

**Theorem 1** *For all $\beta > 0$, under the logistic bandit setting with logistic function $\phi_\beta(x)$, let the quantized parameter $\Theta_\varepsilon$ be defined as in Definition 1 for some $\varepsilon > 0$. If there exists $\Gamma > 0$ such that the average expected TS information ratio is bounded, $\frac{1}{T}\sum_{t=1}^T \mathbb{E}[\Gamma_t] \leq \Gamma$, then the TS regret is bounded as*

$$\mathbb{E}[\text{Regret}(T)] \leq \sqrt{\Gamma T \left(H(\Theta_\varepsilon) + \varepsilon\beta T\right)}.$$

**Proof 1** *We start by rewriting the Thompson Sampling expected regret using the information ratio:*

$$\mathbb{E}[\text{Regret}(T)] = \sum_{t=1}^{T} \mathbb{E}[R(A^\star, \Theta) - R(A_t, \Theta)] = \sum_{t=1}^{T} \mathbb{E}\left[\sqrt{\Gamma_t \mathrm{I}_t(A^\star; R(A_t, \Theta), A_t)}\right].$$

*We continue using Jensen's inequality, followed by Cauchy-Schwarz inequality:*

$$\mathbb{E}[\text{Regret}(T)] \le \sum_{t=1}^{T} \sqrt{\mathbb{E}[\Gamma_t]\mathrm{I}(A^\star; R(A_t, \Theta), A_t | H^t)} \le \sqrt{\Gamma T \sum_{t=1}^{T} \mathrm{I}(A^\star; R(A_t, \Theta), A_t | H^t)},$$

*where in the last inequality, we used that $\sum_{t=1}^{T} \mathbb{E}_t[\Gamma_t] \le \Gamma T$. Applying the chain rule (Yury Polyanskiy, 2022, Theorem 3.7.b) we can decompose the mutual information as*

$$\begin{aligned}
\mathrm{I}(A^\star; R(A_t, \Theta), A_t | H^t) &= \mathrm{I}(A^\star; A_t | H^t) + \mathrm{I}_t(A^\star; R(A_t, \Theta) | H^t, A_t) \\
&= \mathrm{I}(\pi_\star(\Theta); R(A_t, \Theta) | H^t, A_t) \\
&= \mathrm{I}(\Theta; R(A_t, \Theta) | H^t, A_t),
\end{aligned}$$

*where we used the fact that the mutual information $\mathrm{I}_t(A^\star; A_t | H^t) = 0$ as the optimal action $A^\star$ and the Thompson Sampling action $A_t$ are independent conditioned on the history $H^t$ and the last equality follows from (Yury Polyanskiy, 2022, Theorem 3.2.d) as $\pi_\star$ is a one-to-one mapping.*

*Let $f_{R_t | H^t, A_t, \Theta}$ denote the probability density of $R_t$ conditioned on $H^t, A_t, \Theta$ and $f_{R_t | H^t, A_t}$ denote the probability density on $H^t, A_t$. Then, the mutual information terms can be written as*

$$\mathrm{I}(\Theta; R_t | H^t, A_t) = \mathbb{E}\left[\log \frac{f_{R_t | H^t, A_t, \Theta}(R_t)}{f_{R_t | H^t, A_t}(R_t)}\right].$$

*We let the set $\mathcal{O}_\varepsilon$ be an $\varepsilon$-net for $(\mathcal{O}, \rho)$ with associated projection mapping $q : \mathcal{O} \to \mathcal{O}_\varepsilon$. Similarly to the proof of (Neu et al., 2021, Theorem 2), we note that the mutual information can be written as*

$$\mathbb{E}\left[\int_{\mathcal{O}} f_{\Theta | R_t, H^t, A_t}(\theta)\left(\log \frac{f_{R_t | A_t, \Theta=\theta}(R_t)}{f_{R_t | A_t, \Theta=q(\theta)}(R_t)} + \log \frac{f_{R_t | H^t, A_t, \Theta=q(\theta)}(R_t)}{f_{R_t | H^t, A_t}(R_t)}\right) d\theta\right], \quad (1)$$

*since $f_{R_t | H^t, A_t, \Theta} = f_{R_t | A_t, \Theta}$ almost surely by the conditional Markov chain $R_t - A_t - H_t \mid \Theta$.*

*Since the derivative of $\log(\phi_\beta(x))$ is $\beta/(1 + \exp(\beta x))$, it is bounded by $\beta$, which makes it $\beta$-Lipschitz. Furthermore, for all $a \in \mathcal{A}$ and $\theta \in \mathcal{O}$, the inner product $\langle a, \theta \rangle \le 1$, implying that $\log(f_{R_t | A_t, \Theta=\theta}(1))$ is also $\beta$-Lipschitz with respect to $\theta$. Similarly, the derivative of $\log(1 - \phi_\beta(x))$, given by $|\frac{d}{dx}\log(1 - \phi_\beta(x))| = |\beta \exp(\beta x)/(1 + \exp(\beta x))|$, is bounded by $\beta$, making it $\beta$-Lipschitz as well. Consequently, $\log(f_{R_t | A_t, \Theta=\theta}(0))$ is $\beta$-Lipschitz with respect to $\theta$. Thus, we conclude that: $|\log f_{R_t | A_t, \Theta=\theta}(R_t) - \log f_{R_t | A_t, \Theta=q(\theta)}(R_t)| \le \beta\rho(\theta, q(\theta)) \le \beta\varepsilon$.*

*After defining the random variable $\Theta_\varepsilon := q(\Theta)$, the second term in (1) is equal to $\mathrm{I}(\Theta_\varepsilon; R_t | H^t, A_t)$. Summing the $T$ mutual information $\mathrm{I}(\Theta_\varepsilon; R_t | H^t, A_t)$ and applying the chain rule, we obtain*

$$\mathbb{E}[\text{Regret}(T)] \le \sqrt{\Gamma T \left(\mathrm{I}(\Theta_\varepsilon; H^T) + \varepsilon\beta T\right)}.$$

*Finally, we upper bound $\mathrm{I}(\Theta_\varepsilon; H^T)$ by the entropy $\mathrm{H}(\Theta_\varepsilon)$ to obtain the claimed result.*

In the following, we present our main proposition, a bound on the information ratio of Thompson Sampling that depends only on the problem dimension $d$ and the minimax alignment constant $\alpha$.

**Proposition 1** *For all $\beta > 0$, and for all $\mathcal{A}, \mathcal{O} \subseteq \mathbf{B}_d(0, 1)$ with $\alpha$ their minimax alignment constant, under the logistic bandit setting with logistic function $\phi_\beta(x)$, the information ratio of Thompson Sampling is bounded as $\Gamma_t \le \frac{9}{2}d\alpha^{-2}$.*

At a high level, our proof consists of three parts: a lower bound on the conditional mutual information, $\mathrm{I}_t(A^\star; R(A_t, \Theta), A_t)$, an upper bound on the squared expected regret at time $t$, $\mathbb{E}_t[R(A^\star, \Theta) - R(A_t, \Theta)]^2$, and an upper bound on a ratio of expected variances by the study of the limit case $\beta \to \infty$. The key techniques for the proof of Proposition 1 are presented in Section 5.

By combining Proposition 1 with Theorem 1, we arrive at our main result: a bound on the expected Thompson Sampling regret that scales as $O(d/\alpha\sqrt{T\log(\beta T/d)})$. To the best of our knowledge, this is the first regret bound for logistic bandits that scales only logarithmically with the logistic function's parameter $\beta$ while remaining independent of the number of actions.

**Theorem 2** *For all $\beta > 0$, and for all $\mathcal{A}, \mathcal{O} \subseteq \mathbf{B}_d(0,1)$ with $\alpha$ their minimax alignment constant, under the logistic bandit setting with logistic function $\phi_\beta(x)$, the TS regret is bounded as*

$$\mathbb{E}[\text{Regret}(T)] \leq 3d/\alpha\sqrt{T\log\left(\sqrt{3 + \frac{6\beta T}{d}}\right)}.$$

**Proof 2** *After combining Theorem 1 with Proposition 1, we upper bound the entropy $\mathrm{H}(\Theta_\varepsilon)$ by the cardinality of the $\varepsilon$-net to get a regret bound of $3/\alpha\sqrt{dT/2\left(\log(|\Theta_\varepsilon|) + \varepsilon\beta T\right)}$. To define $\Theta_\varepsilon$, we can set $\mathcal{O}_\varepsilon$ as the $\varepsilon$-net of smallest cardinality. As the parameter space $\mathcal{O}$ is within the Euclidean unit ball, we can use Lemma 7 to control the covering number as $\log(|\Theta_\varepsilon|) \leq d\log(1 + 2/\varepsilon)$ and upper bound the Thompson Sampling regret as*

$$\mathbb{E}[\text{Regret}(T)] \leq 3/\alpha\sqrt{dT/2\left(d\log\left(1 + \frac{2}{\varepsilon}\right) + \varepsilon\beta T\right)}.$$

*Finally, setting $\varepsilon = d/(\beta T)$ and rearranging terms inside the logarithm yields the desired result.*

Importantly, the above theorem does not depend on the fragility dimension $\eta$, in contrast to the results of Dong et al. (2019). This distinction is significant because, except in the case where $\alpha = 1$, the fragility dimension can grow exponentially with the dimension $d$. We can verify that our result, due to its logarithmic dependence on $\beta$, is compatible with Dong et al. (2019, Proposition 11), which shows that there exist logistic bandit problems for which no algorithm can achieve a Bayesian regret uniformly bounded by $f(\alpha)p(d)T^{1-\epsilon}$, for some function $f$, polynomial $p$, and $\epsilon > 0$.

The next two corollaries present cases where the dependence on the minimax alignment constant $\alpha$ can be removed. The case in Corollary 2 is particularly relevant for applications where the action set can be treated as a design parameter, and where constructing large action spaces is not prohibitive. We illustrate the improvement of Corollary 1 over previous works through numerical experiments on a synthetic logistic bandit problem. The results are presented in Appendix D.

**Corollary 1** *For all $\beta > 0$, under the logistic bandit setting with logistic function $\phi_\beta(x)$, let $\mathcal{A} \subseteq \mathbf{B}_d(0,1)$ and $\mathcal{O} \subseteq \mathbf{S}_d(0,1)$ be such that $\mathcal{O} \subseteq \mathcal{A}$. Then the TS regret is bounded as*

$$\mathbb{E}[\text{Regret}(T)] \leq 3d\sqrt{T\log\left(\sqrt{3 + \frac{6\beta T}{d}}\right)}.$$

**Proof 3** *If $\mathcal{O} \subseteq \mathbf{S}_d(0,1)$ and the action space $\mathcal{O} \subseteq \mathcal{A}$, then for each $\theta \in \mathcal{O}$, there exists an action $a \in \mathcal{A}$ such that $a = \theta$ and $\langle a, \theta \rangle = 1$, implying $\alpha = 1$. Using this and Theorem 2 concludes the proof.*

**Corollary 2** *For all $\beta > 0$, under the logistic bandit setting with logistic function $\phi_\beta(x)$, there exists an action space $\mathcal{A}$ with $|\mathcal{A}| \leq 2d \cdot 3^{d-1}$ such that for any $\mathcal{O} \subseteq \mathbf{S}_d(0,1)$, the TS regret is bounded as*

$$\mathbb{E}[\text{Regret}(T)] \leq 6d\sqrt{T\log\left(\sqrt{3 + \frac{6\beta T}{d}}\right)}.$$

**Proof 4** *Starting from Theorem 2, we have to construct $\mathcal{A}$ such that its minimimax alignment constant $\alpha$ is greater or equal to $\frac{1}{2}$ for any $\mathcal{O} \subseteq \mathbf{S}_d(0,1)$. This is satisfied if $\mathcal{A}$ is a $\frac{1}{2}$-net for $\mathbf{S}_d(0,1)$. Setting $\mathcal{A}$ as the $\frac{1}{2}$-net of minimal cardinality, from Lemma 8, we have that $|\mathcal{A}| \leq 2d \cdot 3^{d-1}$.*

## 5 ANALYSIS

This section presents the key ideas of the proofs of our main proposition, Proposition 1. For the sake of clarity, we present here our results for the particular setting of Corollary 1, which ensures $\alpha = 1$. We prove how to extend those results to general spaces in Appendix C. Our proof can be divided into three key components: establishing a lower bound on the mutual information (Section 5.1), deriving an upper bound on the squared expected regret (Section 5.2), and obtaining an upper bound on a ratio of expected variances by analyzing the limit case as $\beta \to \infty$ (Section 5.3). To alleviate the notations, we will omit the subscript $t$ for the rest of the section.

A crucial quantity in our analysis is the expected variance of the reward probability conditioned on the sampled action, expressed as $\mathbb{E}[\mathbb{V}[\phi_\beta(\langle \hat{A}, \Theta \rangle)|\hat{A}]]$. It appears in the lower bound on mutual information and a related quantity is used to upper bound the squared expected regret. Intuitively, when the variance of reward probability is high, the agent is exploring new actions, gathering information about $\Theta$ but suffering regret. Conversely, if the variance of reward probability is low, it indicates that the agent has already identified near-optimal actions and is exploiting this knowledge.

Under the logistic bandit setting with logistic function $\phi_\beta$, the reward $R(A_t, \Theta)$ is given by a Bernoulli random variable with associated probability $\phi_\beta(\langle A_t, \Theta \rangle)$. We will denote it by $\mathrm{Bern}(\phi_\beta(\langle A_t, \Theta \rangle))$ to make the setting more explicit. With this notation, we rewrite now the information ratio as:

$$\Gamma = \frac{\mathbb{E}[\mathrm{Bern}(\phi_\beta(\langle A^\star, \Theta \rangle)) - \mathrm{Bern}(\phi_\beta(\langle \hat{A}, \Theta \rangle))]^2}{\mathrm{I}(A^\star; \mathrm{Bern}(\phi_\beta(\langle \hat{A}, \Theta \rangle)), \hat{A})}.$$

### 5.1 LOWER BOUNDING THE MUTUAL INFORMATION

We start by stating a general lemma that relates the mutual information between a random variable $U$ and a Bernoulli random variable with probability $U$ to the variance of the random variable $U$.

**Lemma 1** *Let $U$ be a random variable taking values in $[0, 1]$ and $\mathrm{Bern}(U)$ be a Bernoulli random variable with probability $U$. Then it holds that,*

$$\mathrm{I}(U; \mathrm{Bern}(U)) \geq 2\mathbb{V}(U).$$

**Sketch of proof** *Our proof of Lemma 1 uses the decomposition of mutual information as a difference of entropy and the Taylor expansion of the binary entropy function. It is presented in Appendix A.*

Equipped with Lemma 1, we can prove that the mutual information $\mathrm{I}(A^\star; \mathrm{Bern}(\phi_\beta(\langle \hat{A}, \Theta \rangle)), \hat{A})$ can be lower bounded by the expected variance of reward probability $\mathbb{E}[\mathbb{V}[\phi_\beta(\langle \hat{A}, \Theta \rangle)|\hat{A}]]$.

**Lemma 2** *Let the logistic function be $\phi_\beta(x)$, then, it holds that*

$$\mathrm{I}(A^\star; \mathrm{Bern}(\phi_\beta(\langle \hat{A}, \Theta \rangle)), \hat{A}) \geq 2\mathbb{E}\left[\mathbb{V}\left[\phi_\beta(\langle \hat{A}, \Theta \rangle) \mid \hat{\Theta}\right]\right].$$

**Proof 5** *We start the proof by applying the chain rule. It comes that*

$$
\begin{aligned}
\mathrm{I}(\pi_\star(\Theta); \pi_\star(\hat{\Theta}), \mathrm{Bern}(\phi_\beta(\langle \hat{A}, \Theta \rangle))) &\overset{(i)}{=} \mathrm{I}(\Theta; \hat{\Theta}, \mathrm{Bern}(\phi_\beta(\langle \hat{A}, \Theta \rangle))) \\
&\overset{(j)}{=} \mathrm{I}(\Theta; \hat{\Theta}) + \mathrm{I}(\Theta; \mathrm{Bern}(\phi_\beta(\langle \hat{A}, \Theta \rangle)) \mid \hat{\Theta}) \\
&\overset{(k)}{=} \mathrm{I}(\Theta; \mathrm{Bern}(\phi_\beta(\langle \hat{A}, \Theta \rangle)) \mid \hat{\Theta}) \\
&\overset{(l)}{=} \mathbb{E}[\mathrm{I}(\phi_\beta(\langle \hat{A}, \Theta \rangle); \mathrm{Bern}(\phi_\beta(\langle \hat{A}, \Theta \rangle))) \mid \hat{\Theta} = \theta],
\end{aligned}
$$

*where (i) follows as $\pi_\star$ is a one-to-one mapping; (j) follows from the chain-rule; (k) follows as $\Theta$ and $\hat{\Theta}$ are independent conditioned on the history; and (l) is obtained using the fact that $\phi_\beta(\langle a, \theta \rangle)$ is a one-to-one mapping conditioned on $\hat{\Theta} = \theta$. Finally, applying Lemma 1 yields the desired result.*

## 5.2 UPPER BOUNDING THE SQUARED EXPECTED REGRET

This part of the proof takes inspiration from the proof techniques of (Dong & Van Roy, 2018, Proposition 15) and similar to them, the two following lemmata will be of importance for our analysis.

**Lemma 3 ((Dong & Van Roy, 2018, Lemma 16))** *Let $U, V$ be random vectors in $\mathbb{R}^d$, and let $\tilde{U}, \tilde{V}$ be independent random variables with distributions equal respectively to the marginals of $U, V$, then*

$$\mathbb{E}\left[\left(U^\top V\right)\right]^2 \leq d \cdot \mathbb{E}\left[\left(\tilde{U}^\top \tilde{V}\right)^2\right].$$

**Lemma 4 ((Dong & Van Roy, 2018, Lemma 18))** *Let $f : \mathbb{R}_+ \to \mathbb{R}_+$ be such that $f(0) \geq 0$ and $f(\zeta)/\zeta$ is non-decreasing over $\zeta \geq 0$. Then, for any non-negative random variable $U$, there is*

$$\frac{\mathbb{E}[f(U)]^2}{\mathbb{E}[U]^2} \leq \frac{\mathbb{V}[f(U)]}{\mathbb{V}[U]}.$$

When $\mathcal{O} \subseteq \mathbf{S}_d(0,1)$ and the action space $\mathcal{O} \subseteq \mathcal{A}$, then for each $\theta \in \mathcal{O}$, there exists an action $a \in \mathcal{A}$ such that $a = \theta$ and $\langle a, \theta \rangle = 1$. This implies that $\phi_\beta(\langle A^\star, \Theta \rangle) - \phi_\beta(\langle \hat{A}, \Theta \rangle) = \phi_\beta(1) - \phi_\beta(\langle \hat{A}, \Theta \rangle)$. To simplify notation, we define $\psi_\beta(x) := \phi_\beta(1) - \phi_\beta(1 - x)$, which relates the difference between the optimal action $A^\star$ and the sampled action $\hat{A}$ to their corresponding reward differences. Specifically, we have $\psi_\beta(1 - \langle \hat{A}, \Theta \rangle) = \psi_\beta(\langle A^\star - \hat{A}, \Theta \rangle) = \phi_\beta(\langle A^\star, \Theta \rangle) - \phi_\beta(\langle \hat{A}, \Theta \rangle)$.

The function $\psi_\beta(x)$ meets the first two conditions from Lemma 4. Applied to the difference of inner products $\langle A^\star, \Theta \rangle - \langle \hat{A}, \Theta \rangle$, it maps the interval $[0, 2]$ to $[0, 1]$, and satisfies $\psi_\beta(0) = \phi_\beta(1) - \phi_\beta(1 - 0) = 0$. However, it does not meet the third condition, as $\psi_\beta(x)/x$ initially increases, reaches a maximum between 1 and 2, and then decreases (see Remark 1 ). To address this issue, we introduce a modified function, referred to as the *logistic surrogate*, which serves as the tightest upper bound on $\psi_\beta(x)$ that satisfies the final requirement from Lemma 4.

**Definition 2 (Logistic surrogate)** *We construct the* logistic surrogate *function $\varphi_\beta(x)$ as the tightest upper bound on $\psi_\beta(x)$ such that $\varphi_\beta(x)/x$ is non-decreasing over $x \geq 0$.*

*Namely, let $\delta_\beta = \arg\max_{x \in [0,2]} \frac{\psi_\beta(x)}{x}$, we define the function $\varphi_\beta$ as*

$$\varphi_\beta(x) = \begin{cases} \psi_\beta(x) & x \in [0, \delta_\beta] \\ \psi_\beta(\delta_\beta) + (x - \delta_\beta) \cdot \psi_\beta(\delta_\beta)/\delta_\beta & x \in ]\delta_\beta, 2] \end{cases}.$$

We are now equipped to state and prove an upper bound on the squared expected regret.

**Lemma 5** *Let the logistic surrogate be defined as in Definition 2. Then, it holds that*

$$\mathbb{E}[\mathrm{Bern}(\phi_\beta(\langle A^\star, \Theta \rangle)) - \mathrm{Bern}(\phi_\beta(\langle \hat{A}, \Theta \rangle))]^2 \leq d \cdot \mathbb{E}\left[\mathbb{V}\left[\varphi_\beta\left(1 - \langle \hat{A}, \Theta \rangle\right) \mid \hat{\Theta}\right]\right].$$

**Proof 6** *By integrating over the randomness of the Bernoulli outcome, the squared expected regret can be expressed as $\mathbb{E}[(\phi_\beta(\langle A^\star, \Theta \rangle) - \phi_\beta(\langle \hat{A}, \Theta \rangle))]^2 = \mathbb{E}[\psi_\beta(1 - \langle \hat{A}, \Theta \rangle)]^2$. Since by definition $\varphi_\beta(x) \geq \psi_\beta(x)$, we have $\mathbb{E}[\psi_\beta(1 - \langle \hat{A}, \Theta \rangle)]^2 \leq \mathbb{E}[\mathbb{E}[\varphi_\beta(1 - \langle \hat{A}, \Theta \rangle)|\hat{\Theta}]]^2$.*

*We now apply Lemma 4 on $\mathbb{E}[\varphi_\beta(1 - \langle \hat{A}, \Theta \rangle)|\hat{\Theta}]$. It comes that*

$$\mathbb{E}[\mathbb{E}[\varphi_\beta(1 - \langle \hat{A}, \Theta \rangle)|\hat{\Theta}]]^2 \leq \mathbb{E}\left[\underbrace{\sqrt{\frac{\mathbb{V}\left[\varphi_\beta\left(1 - \langle \hat{A}, \Theta \rangle\right) \mid \hat{\Theta}\right]}{\mathbb{V}\left[1 - \langle \hat{A}, \Theta \rangle \mid \hat{\Theta}\right]}}}_{:=U(\hat{\Theta})} \mathbb{E}\left[1 - \langle \hat{A}, \Theta \rangle \mid \hat{\Theta}\right]\right]^2$$

$$= \mathbb{E}\left[U(\hat{\Theta})\langle \hat{A}, \hat{\Theta} \rangle - \langle \hat{A}, \Theta \rangle\right]^2 = \mathbb{E}\left[\langle U(\hat{\Theta})\hat{A}, \Theta - \hat{\Theta} \rangle\right]^2.$$

*We use Lemma 3 with $U = U(\hat{\Theta})\hat{A}$ and $V = \Theta - \hat{\Theta}$ and rearrange terms to obtain the claimed result:*

$$\mathbb{E}\left[\langle U(\hat{\Theta})\hat{A}, \Theta - \hat{\Theta}\rangle\right]^2 \le d \cdot \mathbb{E}\left[\left(\langle U(\hat{\Theta})\hat{A}, \Theta - \tilde{\Theta}\rangle\right)^2\right] = d \cdot \mathbb{E}\left[U(\hat{\Theta})^2 \mathbb{E}\left[\langle \hat{A}, \Theta - \tilde{\Theta}\rangle^2 | \hat{\Theta}\right]\right]$$

$$= d \cdot \mathbb{E}\left[\frac{\mathbb{V}\left[\varphi_\beta\left(1 - \langle \hat{A}, \Theta\rangle\right) | \hat{\Theta}\right]}{\mathbb{V}\left[1 - \langle \hat{A}, \Theta\rangle | \hat{\Theta}\right]} \mathbb{V}\left[\langle \hat{A}, \Theta\rangle | \hat{\Theta}\right]\right]$$

$$= d \cdot \mathbb{E}\left[\mathbb{V}\left[\varphi_\beta\left(1 - \langle \hat{A}, \Theta\rangle\right) | \hat{\Theta}\right]\right].$$

Combining Lemma 2 and Lemma 5, we get that the information ration $\Gamma$ is bounded by

$$\Gamma \le d/2 \cdot \frac{\mathbb{E}\left[\mathbb{V}\left[\varphi_\beta\left(1 - \langle \hat{A}, \Theta\rangle\right) | \hat{\Theta}\right]\right]}{\mathbb{E}\left[\mathbb{V}\left[\psi_\beta\left(1 - \langle \hat{A}, \Theta\rangle\right) | \hat{\Theta}\right]\right]},$$

where we use the fact that $\mathbb{V}\left[\phi_\beta(\langle \hat{A}, \Theta\rangle) | \hat{\Theta}\right] = \mathbb{V}\left[\psi_\beta(1 - \langle \hat{A}, \Theta\rangle) | \hat{\Theta}\right]$ by the definition of $\psi_\beta$. The next part of the proof takes care of controlling the ratio of expected variances over $\varphi_\beta$ and $\psi_\beta$.

### 5.3 BOUNDING THE RATIO OF EXPECTED VARIANCES OVER THE FUNCTIONS $\varphi_\beta$ AND $\psi_\beta$

By definition, the function $\psi_\beta$ and its surrogate $\varphi_\beta$ are equal for $x \in [0, \delta_\beta]$ and then diverge linearly at a rate of $\psi_\beta(\delta_\beta)/\delta_\beta$. We observe, in Remark 1, that $\delta_\beta$ is a decreasing function of $\beta$ and that the slope $\psi_\beta(\delta_\beta)/\delta_\beta$ strictly increases with $\beta$. This observation suggests that studying the case $\beta \to \infty$ could provide a general upper bound. Indeed, taking the limit case $\beta \to \infty$, the domain where the two functions differ is maximized, and the rate at which they differ is the largest.

We show in Lemma 9, presented in Appendix B, that under some simple preliminary transformations, increasing the value of $\beta$ leads to a larger ratio of expected variances, and therefore, the case $\beta$ tending to $\infty$ can serve to derive general upper bounds. Quite satisfyingly, this limit case provides a lot of simplifications. We will prove in Lemma 9, that the ratio of expected variance between $\psi_\beta$ and $\varphi_\beta$ can be upper bounded by the ratio of expected variance between $\overline{\psi}$ and $\overline{\varphi}$ defined as

$$\overline{\psi}(x) = \begin{cases} 0 & x \in [0, 1] \\ 1 & x \in ]1, 2] \end{cases}, \tag{2}$$

and

$$\overline{\varphi}(x) = \begin{cases} 0 & x \in [0, 1] \\ 1 + 2(x - 1) & x \in ]1, 2] \end{cases}. \tag{3}$$

**Lemma 6** *Let $\overline{\psi}$ and $\overline{\varphi}$ be defined respectively in (2) and (3). Then, it holds that*

$$\frac{\mathbb{E}\left[\mathbb{V}\left[\overline{\varphi}\left(1 - \langle \hat{A}, \Theta\rangle\right) | \hat{\Theta}\right]\right]}{\mathbb{E}\left[\mathbb{V}\left[\overline{\psi}\left(1 - \langle \hat{A}, \Theta\rangle\right) | \hat{\Theta}\right]\right]} \le 9.$$

**Sketch of proof** *Analyzing the function $\overline{\psi}$, we note that $\mathbb{E}[\mathbb{V}[\overline{\psi}(1 - \langle \hat{A}, \Theta\rangle) | \hat{\Theta}]]$ is equal to the expected variance of a Bernoulli random variable with probability given by $Q(\hat{A}) = \mathbb{E}[I(\langle \hat{A}, \Theta\rangle)]$ with $I(\langle \hat{A}, \Theta\rangle) = \mathbb{1}_{\{\langle \hat{A}, \Theta\rangle < 0\}}$. The expected variance can therefore be written as $\mathbb{E}[I(\langle \hat{A}, \Theta\rangle)^2] - \mathbb{E}[\mathbb{E}[I(\langle \hat{A}, \Theta\rangle)]^2]$ where in the second term, the outer expectation is on $\hat{A}$, and the inner expectation is on $\Theta$. After some rearranging of the terms, we can write $\mathbb{E}[\mathbb{V}[\overline{\varphi}(1 - \langle \hat{A}, \Theta\rangle) | \hat{\Theta}]]$ as $\mathbb{E}[I(\langle \hat{A}, \Theta\rangle)(1 - 2\langle \hat{A}, \Theta\rangle)^2] - \mathbb{E}[\mathbb{E}[I(\langle \hat{A}, \Theta\rangle)(1 - 2\langle \hat{A}, \Theta\rangle)]^2]$ where again for the second term, the outer expectation is on $\hat{A}$, and the inner expectation is on $\Theta$. Taking the supremum over the possible values of $(1 - 2\langle a, \theta\rangle) \in [-1, 3]$ concludes the proof.*

## 6 CONCLUSION AND FUTURE WORK

In this work, we analyzed the performance of the Thompson Sampling algorithm for logistic bandit problems, focusing on settings where both the action and parameter spaces lie within the $d$-dimensional unit ball. Using an information-theoretic framework, we provided a refined analysis of the information ratio, a key statistic that captures the trade-off between exploration and exploitation in logistic bandits. Our main result established that the information ratio is bounded by $\frac{9}{2}d\alpha^{-2}$, where $\alpha$ is a minimax alignment constant between the action and parameter spaces. Importantly, this bound is independent of the logistic function's slope parameter, $\beta$.

Building on this, we derived a regret bound of $O(d/\alpha\sqrt{T\log(\beta T/d)})$, which scales only logarithmically with $\beta$, representing a significant improvement over prior results. To the best of our knowledge, this is the first regret bound for logistic bandits that achieves logarithmic dependence on $\beta$ while remaining independent of the action set's cardinality. Importantly, our results do not depend on the fragility dimension $\eta$, unlike those of Dong et al. (2019). This distinction is significant because, except in cases where $\alpha = 1$, the fragility dimension can grow exponentially with the dimension $d$. Finally, we presented specific settings where the dependence on $\alpha$ can be controlled. For instance, when the action space fully encompasses the parameter space, the regret of Thompson Sampling scales as $\tilde{O}(d\sqrt{T})$.

A promising direction for future work is to extend our analysis to the broader class of generalized linear bandits. The key properties of the logistic function that we leverage could potentially extend to other link functions, such as those from exponential family models or Poisson regression.

Another exciting avenue is exploring whether our approach can be applied to the *optimistic information directed sampling* algorithm introduced by (Neu et al., 2024), with the goal of deriving frequentist regret bounds for logistic bandits that scale logarithmically with the parameter $\beta$. Extending our analysis to the frequentist setting would represent a significant advancement in the field.

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

APPENDIX

The appendix is organized as follows:

- Appendix A introduces some useful lemmata for our proofs.
- Appendix B provides the formal proof to control the ratio of expected variances between the functions $\varphi_\beta$ and $\psi_\beta$.
- Appendix C presents the techniques to extend our analysis of the Thompson Sampling information ratio to general action and parameter spaces.
- Appendix D illustrates the improvement of our bounds compared to previous regret guarantees through numerical experiments.
- Appendix E elaborates on the gaps in the previous literature mentioned in Section 1.
- Appendix F rigorously explains the construction of the mapping $\pi_\star$.

## A  USEFUL LEMMATA

**Lemma 1** *Let $U$ be a random variable taking values in $[0,1]$ and $\mathrm{Bern}(U)$ be a Bernoulli random variable with probability $U$. Then it holds that,*

$$\mathrm{I}(U; \mathrm{Bern}(U)) \geq 2\mathbb{V}(U).$$

**Proof 7** *The proof uses the decomposition of mutual information as a difference of entropy and the Taylor expansion of the binary entropy function. Using proposition (Yury Polyanskiy, 2022)[Theorem 3.4.d], we can decompose the mutual information between $U$ and $\mathrm{Bern}(U)$ as*

$$\mathrm{I}(U; \mathrm{Bern}(U)) = h(\mathrm{Bern}(U)) - h(\mathrm{Bern}(U)|U).$$

*Following (Duchi, 2016)[Example 2.2] notation, we define $h_2(p) := -p\log(p) - (1-p)\log(1-p)$ for $p \in [0,1]$. We note that we can rewrite the mutual information as*

$$\mathrm{I}(U; \mathrm{Bern}(U)) = h_2(\mathbb{E}[U]) - \mathbb{E}[h_2(U)]. \tag{4}$$

*From a Taylor expansion of $h_2(x)$ we have that*

$$h_2(x) = h_2(p) + (x-p)h_2'(p) + \frac{1}{2}(x-p)^2 h_2''(\xi),$$

*for some $\xi \in (0,1)$ as $h_2''$ is continuous on the interval $[0,1]$. We compute the second derivative of $h_2$ and get $h_2''(\xi) = -\frac{1}{\xi(1-\xi)}$ for $\xi \in (0,1)$. This function is concave and maximal at $\xi = 1/2$, where it takes the value $h_2''(1/2) = -4$. We then have that for all $x \in [0,1]$ and all $p \in [0,1]$,*

$$h_2(x) \leq h_2(p) + (x-p)h_2'(p) - 2(x-p)^2.$$

*Using this fact for $x = U$ and $p = \mathbb{E}[U]$, we have that*

$$h_2(U) \leq h_2(\mathbb{E}[U]) + (U - \mathbb{E}[U])h_2'(\mathbb{E}[U]) - 2(U - \mathbb{E}[U])^2.$$

*Applying the last inequality to the second term in (4), it comes that*

$$\mathrm{I}(U; \mathrm{Bern}(U)) \geq \mathbb{E}\left[h_2(\mathbb{E}[U]) - h_2(\mathbb{E}[U]) - (U - \mathbb{E}[U])h_2'(\mathbb{E}[U]) + 2(U - \mathbb{E}[U])^2\right].$$

*Finally, simplifying terms and taking the expectation gives the desired result.*

The two following lemmata are be particularly useful to control the covering number in Euclidean balls and spheres.

**Lemma 7 ((van Handel, 2016, Lemma 5.13))** *Let $\mathbf{B}_d(0,1)$ denote the $d$-dimensional closed Euclidean unit ball. We have $|\mathcal{N}(\mathbf{B}_d(0,1), ||\cdot||_2, \varepsilon)| = 1$ for $\varepsilon \geq 1$ and for $0 < \varepsilon < 1$, we have*

$$\left(\frac{1}{\varepsilon}\right)^d \leq |\mathcal{N}(\mathbf{B}_d(0,1), ||\cdot||_2, \varepsilon)| \leq \left(1 + \frac{2}{\varepsilon}\right)^d.$$

**Lemma 8 ((Yury Polyanskiy, 2022, Corollary 27.4))** *Let $\mathbf{S}_d(0,1)$ denote the $d$-dimensional Euclidean unit sphere. We have $|\mathcal{N}(\mathbf{S}_d(0,1), ||\cdot||_2, \varepsilon) = 1$ for $\varepsilon \geq 1$ and for $0 < \varepsilon < 1$, we have*

$$\left(\frac{1}{2\varepsilon}\right)^{d-1} \leq |\mathcal{N}(\mathbf{S}_d(0,1), ||\cdot||_2, \varepsilon)| \leq 2d\left(1 + \frac{1}{\varepsilon}\right)^{d-1}.$$

## B  BOUNDING THE RATIO OF EXPECTED VARIANCES OVER THE FUNCTIONS $\varphi_\beta$ AND $\psi_\beta$

**Lemma 6** *Let $\overline{\psi}$ and $\overline{\varphi}$ be defined respectively in (2) and (3). Then, it holds that*

$$\frac{\mathbb{E}\left[\mathbb{V}\left[\overline{\varphi}\left(1-\langle\hat{A},\Theta\rangle\right)\mid\hat{\Theta}\right]\right]}{\mathbb{E}\left[\mathbb{V}\left[\overline{\psi}\left(1-\langle\hat{A},\Theta\rangle\right)\mid\hat{\Theta}\right]\right]}\leq 9.$$

**Proof 8** *We start by analyzing $\mathbb{E}\left[\mathbb{V}\left[\overline{\psi}\left(1-\langle\hat{A},\Theta\rangle\right)\mid\hat{\Theta}\right]\right]$. We note that $\overline{\psi}\left(1-\langle\hat{A},\Theta\rangle\right)$ is equal to 1 if $\langle\hat{A},\Theta\rangle < 0$ and is equal to 0 otherwise. To distinguish those two cases, we introduce the notation $I(\langle\hat{A},\Theta\rangle) := \mathbb{1}_{\{\langle\hat{A},\Theta\rangle<0\}}$. We observe that $\mathbb{E}\left[\mathbb{V}\left[\overline{\psi}\left(1-\langle\hat{A},\Theta\rangle\right)\mid\hat{\Theta}\right]\right]$ is equal to the expected variance of a Bernoulli random variable with probability given by $Q(\hat{A}) := \mathbb{E}[I(\langle\hat{A},\Theta\rangle)]$ and can therefore be written as*

$$\mathbb{E}\left[\mathbb{V}\left[\overline{\psi}\left(1-\langle\hat{A},\Theta\rangle\right)\mid\hat{\Theta}\right]\right] = \mathbb{E}[Q(\hat{A})(1-Q(\hat{A}))].$$

*The last part of the proof concerns $\mathbb{E}\left[\mathbb{V}\left[\overline{\varphi}\left(1-\langle\hat{A},\Theta\rangle\right)\mid\hat{\Theta}\right]\right]$. Similarly, we can distinguish between two cases: either $\langle\hat{A},\Theta\rangle > 0$ and $\overline{\varphi}(1-\langle\hat{A},\Theta\rangle) = 0$, or $\langle\hat{A},\Theta\rangle < 0$ and $\overline{\varphi}(1-\langle\hat{A},\Theta\rangle) = 1 - 2\langle\hat{A},\Theta\rangle$. Introducing the notation $G(\hat{A}) := \mathbb{E}[I(\langle\hat{A},\Theta\rangle)\langle\hat{A},\Theta\rangle]$, we can write*

$$\mathbb{E}\left[\mathbb{V}\left[\overline{\varphi}\left(1-\langle\hat{A},\Theta\rangle\right)\mid\hat{\Theta}\right]\right] = \mathbb{E}\left[\mathbb{E}\left[\left(\overline{\varphi}\left(1-\langle\hat{A},\Theta\rangle\right)-\mathbb{E}\left[\overline{\varphi}\left(1-\langle\hat{A},\Theta\rangle\right)\mid\hat{\Theta}\right]\right)^2\mid\hat{\Theta}\right]\right]$$

$$= \mathbb{E}\left[I(\langle\hat{A},\Theta\rangle)\left(1-2\langle\hat{A},\Theta\rangle-\left(Q(\hat{A})+2G(\hat{A})\right)\right)^2\right]$$

$$+ \mathbb{E}\left[\left(1-I(\langle\hat{A},\Theta\rangle)\right)\left(0-\left(Q(\hat{A})+2G(\hat{A})\right)\right)^2\right].$$

*Distributing the square and simplifying terms, we obtain*

$$\mathbb{E}\left[I(\langle\hat{A},\Theta\rangle)\left(1-2\langle\hat{A},\Theta\rangle\right)^2\right] - 2\mathbb{E}\left[I(\langle\hat{A},\Theta\rangle)\left(1-2\langle\hat{A},\Theta\rangle\right)\left(Q(\hat{A})+2G(\hat{A})\right)\right]$$

$$+ \mathbb{E}\left[I(\langle\hat{A},\Theta\rangle)\left(Q(\hat{A})+2G(\hat{A})\right)^2\right] + \mathbb{E}\left[\left(1-I(\langle\hat{A},\Theta\rangle)\right)\left(Q(\hat{A})+2G(\hat{A})\right)^2\right]$$

$$= \mathbb{E}\left[I(\langle\hat{A},\Theta\rangle)\left(1-2\langle\hat{A},\Theta\rangle\right)^2\right] - \mathbb{E}\left[\left(Q(\hat{A})+2G(\hat{A})\right)^2\right].$$

*To get to the last part of the proof, we rewrite explicitly $Q(\hat{A}) + 2G(\hat{A})$ as $\mathbb{E}\left[I(\langle\hat{A},\Theta\rangle)\left(1-2\langle\hat{A},\Theta\rangle\right)\right]$ and optimize over the values of $(1-2\langle\hat{A},\Theta\rangle)$. It then comes*

$$\mathbb{E}\left[I(\langle\hat{A},\Theta\rangle)\left(1-2\langle\hat{A},\Theta\rangle\right)^2\right] - \mathbb{E}\left[\mathbb{E}\left[I(\langle\hat{A},\Theta\rangle)\left(1-2\langle\hat{A},\Theta\rangle\right)\right]^2\right]$$

$$\leq \sup_{\zeta\in[-1,3]}\mathbb{E}\left[I(\langle\hat{A},\Theta\rangle)\zeta^2\right] - \mathbb{E}\left[\mathbb{E}\left[I(\langle\hat{A},\Theta\rangle)\zeta\right]^2\right] = 9\cdot\mathbb{E}[Q(\hat{A})(1-Q(\hat{A}))],$$

*which concludes the proof.*

**Lemma 9** *Let $\psi_\beta(x) = \phi_\beta(1) - \phi(1-x)$ and the logistic surrogate $\varphi_\beta$ as in Definition 2 and let $\overline{\psi}$ and $\overline{\varphi}$ be defined respectively in (2) and (3). Then, for all $\beta > 0$, it holds that*

$$\frac{\mathbb{E}\left[\mathbb{V}\left[\varphi_\beta\left(1-\langle\hat{A},\Theta\rangle\right)\mid\Theta\right]\right]}{\mathbb{E}\left[\mathbb{V}\left[\psi_\beta\left(1-\langle\hat{A},\Theta\rangle\right)\mid\hat{\Theta}\right]\right]}\leq\frac{\mathbb{E}\left[\mathbb{V}\left[\overline{\varphi}\left(1-\langle\hat{A},\Theta\rangle\right)\mid\hat{\Theta}\right]\right]}{\mathbb{E}\left[\mathbb{V}\left[\overline{\psi}\left(1-\langle\hat{A},\Theta\rangle\right)\mid\hat{\Theta}\right]\right]}.$$

**Proof 9** *Beginning with the ratio of expected variances between $\varphi_\beta$ and $\psi_\beta$, we will apply a series of transformations to the functions $\varphi_\beta$ and $\psi_\beta$, ultimately yielding the functions $\overline{\varphi}$ and $\overline{\psi}$. These transformations are chosen to ensure they can only increase the ratio of expected variances.*

*By definition, the function $\psi_\beta$ and its surrogate $\varphi_\beta$ are identical for $x \in [0, \delta_\beta)$ and then diverge linearly at a rate of $\psi_\beta(\delta_\beta)/\delta_\beta$ on the interval $x \in [\delta_\beta, 2]$. We illustrate this on Figure 3. Focusing on the domain where the two functions coincide, we observe that the transformation $f(x) = \max(x, \psi_\beta(1))$ reduces the expected variance for both $\psi_\beta$ and $\varphi_\beta$. However, since $\psi_\beta(x)$ is less than or equal to $\varphi_\beta(x)$ for all $x \in [0, 2]$, and both functions exceed $\psi_\beta(1)$ on the interval $[1, 2]$, the transformation $f$ proportionally reduces the expected variance of $\psi_\beta$ more than that of $\varphi_\beta$. As a result, the transformation increases the ratio of expected variances between the two functions. As $\psi_\beta$ and $\varphi_\beta$ are strictly increasing functions, the resulting functions, illustrated on Figure 4, can be written as*

$$f(\psi_\beta(x)) = \begin{cases} \psi_\beta(1) & x \in [0, 1] \\ \psi_\beta(x) & x \in \,]1, 2] \end{cases},$$

*and*

$$f(\varphi_\beta(x)) = \begin{cases} \psi_\beta(1) & x \in [0, 1] \\ \varphi_\beta(x) & x \in \,]1, 2] \end{cases}.$$

*The second transformation we apply concerns only the function $f(\psi_\beta(x))$. We will crop all the values larger than $\psi(\delta_\beta)$ by applying the transformation $g(x) = \min(x, \psi(\delta_\beta))$. As $f(\psi_\beta(x))$ is an increasing function, the function $g(f(\psi_\beta(x)))$, illustrated on Figure 5, can be written as*

$$g(f(\psi_\beta(x))) = \begin{cases} \psi_\beta(1) & x \in [0, 1] \\ \psi_\beta(x) & x \in \,]1, \delta_\beta] \\ \psi_\beta(\delta_\beta) & x \in \,]\delta_\beta, 2] \end{cases}.$$

*The transformation $g$ reduces the variance of the function $f(\psi_\beta(x))$ as it both decreases the values of $f(\psi_\beta(x))$ and the derivative of $f(\psi_\beta(x))$ for all $x \in \,]\delta_\beta, 2]$.*

*The third transformation we apply is increasing the value of $\beta$. As $\beta$ increases, the derivative of $f(\varphi_\beta(x))$ increases everywhere,*

$$\frac{d}{dx} f(\varphi_\beta(x)) = \begin{cases} 0 & x \in [0, 1] \\ \frac{\beta \exp(-\beta(1-x))}{(1+\exp(-\beta(1-x)))^2} & x \in \,]1, \delta_\beta] \\ \psi_\beta(\delta_\beta)/\delta_\beta & x \in \,]\delta_\beta, 2] \end{cases}.$$

*and the expected variance of $f(\varphi_\beta)$ increases. Regarding $g(f(\psi_\beta(x)))$, we can show that that for all $x \in [0, 2]$, the ratio $f(\varphi_\beta(x))/g(f(\psi_\beta(x)))$ increases with $\beta$. Indeed, this ratio is equal to 1 for all $x \in [0, \delta_\beta]$ and increases for all $x \in \,]\delta_\beta, 2]$ as*

$$\frac{f(\varphi_\beta(x))}{g(f(\psi_\beta(x)))} = \frac{\varphi_\beta(\delta_\beta) + \varphi_\beta(\delta_\beta)/\delta_\beta \cdot (x - \delta_\beta)}{\varphi_\beta(\delta_\beta)} = 1 + \frac{(x - \delta_\beta)}{\delta_\beta},$$

*and as $\delta_\beta$ is a decreasing function of $\beta$ (see Remark 1), the ratio $(x - \delta_\beta)/\delta_\beta$ is a increasing function of $\beta$ for all $x \in \,]\delta_\beta, 2]$. This fact ensures that the expected variance of $g(f(\psi_\beta(x)))$ cannot increase proportionally more than the expected variance of $f(\varphi_\beta(x))$. We can therefore study the ratio of expected variances between $f(\varphi_\infty)$ and $g(f(\psi_\infty))$.*

*The last operation we apply is merely a convenient shifting and scaling, where we define $h(x) = (x - g(f(\psi_\beta(1))))/(g(f(\psi_\infty(2))) - g(f(\psi_\beta(1))))$. Applied on both $g(f(\psi_\infty))$ and $f(\varphi_\infty)$ these operations do not affect the ratio of expected variances. The resulting functions are illustrated on Figure 6.*

*To express the resulting functions, we have to analyze the function $\psi_\beta(x)$ for $\beta$ tending to infinity for values $x \in \,]1, 2]$.*

*We recall that $\psi_\beta(x) = \phi_\beta(1) - \phi_\beta(1 - x)$ and can equivalently be written as*

$$\psi_\beta(x) = \frac{1}{1 + \exp(-\beta)} - \frac{1}{1 + \exp(-\beta(1 - x))}.$$

*We have to distinguish between three cases for $(x - 1)$: negative, zero, or positive. For values of $x \in ]1, 2]$, we have that $(1 - x) < 0$ and that $\lim_{\beta \to \infty} \psi_\beta(x) = 1$, if $x = 1$, we have that $\lim_{\beta \to \infty} \psi_\beta(x) = 1/2$ and for values of $x \in [0, 1[$, we have that $(1 - x) > 0$ and that $\lim_{\beta \to \infty} \psi_\beta(x) = 0$. We can then write*

$$\psi_\infty(x) = \begin{cases} 0 & x \in [0, 1[ \\ 1/2 & x = 1 \\ 1 & x \in ]1, 2] \end{cases}.$$

*We can now construct the corresponding $\varphi_\infty(x)$. We note that $\frac{\psi_\infty(x)}{x}$ is maximized when taking the limit to $x = 1^+$ from the right: $\lim_{x \to 1^+} \frac{\psi_\infty(x)}{x} = 1$. It comes that $\varphi_\infty(x)$ can be written as*

$$\varphi_\infty(x) = \begin{cases} 0 & x \in [0, 1[ \\ 1/2 & x = 1 \\ 1 + (x - 1) & x \in ]1, 2] \end{cases}.$$

*We denote the resulting functions $h(g(f(\psi_\infty(x))))$ and $h(f(\varphi_\infty(x)))$ respectively as $\overline{\psi}$ and $\overline{\varphi}$. We note that they can be written quite simply as*

$$\overline{\psi}(x) = \begin{cases} 0 & x \in [0, 1] \\ 1 & x \in ]1, 2] \end{cases},$$

*and*

$$\overline{\varphi}(x) = \begin{cases} 0 & x \in [0, 1] \\ 1 + 2(x - 1) & x \in ]1, 2] \end{cases}.$$

**Remark 1** *We illustrate the function $\psi_\beta(x)/x$ on Figure 1 and the behavior of $\delta_\beta$ and $\psi_\beta(\delta_\beta)/\delta_\beta$ for increasing values of $\beta$ on Figure 2. The derivative of the function $\psi_\beta(x)/x$ is given by*

$$\frac{d}{dx}\left(\frac{\psi_\beta(x)}{x}\right) = \frac{1}{x}\left(\frac{d}{dx}\psi_\beta(x) - \frac{\psi_\beta(x)}{x}\right).$$

*We note that it is equal to zero for values of $x \in ]0, 2]$ such that $\frac{d}{dx}\psi_\beta(x) = \frac{\psi_\beta(x)}{x}$. By definition of $\delta_\beta$, we have $\frac{d}{dx}\psi_\beta(\delta_\beta) = \frac{\psi_\beta(\delta_\beta)}{\delta_\beta}$.*

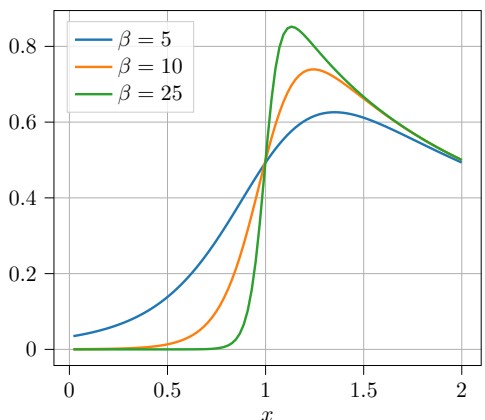

Figure 1: Illustration of the function $\psi_\beta(x)/x$ for different values of $x$. The maximum of the function is attained for $x = \delta_\beta$.

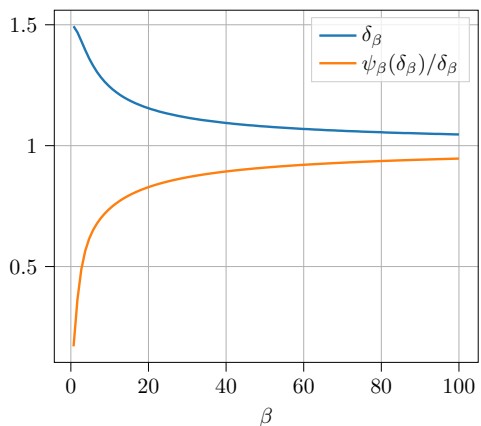

Figure 2: Illustration of $\delta_\beta$ and $\psi(\delta_\beta)/\delta_\beta$ as functions of $\beta$. One can observe that $\delta_\beta$ decreases with $\beta$ while $\psi(\delta_\beta)/\delta_\beta$ increases.

## C    EXTENSION TO GENERAL SPACES

The extend the proof technique of Section 5.2 and Section 5.3, we first need to introduce the *alignment function* $\alpha(\theta) := \langle \pi_\star(\theta), \theta \rangle$. We can define the *extended logistic function* $\psi_\beta(x, \theta) := \phi_\beta(\alpha(\theta)) - \phi_\beta(\alpha(\theta) - x)$ and note that

$$\psi_\beta(\alpha(\hat{\Theta}) - \langle \hat{A}, \Theta \rangle, \hat{\Theta}) = \phi_\beta(\alpha(\hat{\Theta})) - \phi_\beta(\langle \hat{A}, \Theta \rangle) = \phi_\beta(\langle \hat{A}, \hat{\Theta} \rangle) - \phi_\beta(\langle \hat{A}, \Theta \rangle).$$

Integrating the randomness of the Bernoulli process, we can write the expected regret using $\psi_\beta(x, \theta)$:

$$\mathbb{E}[\psi_\beta(\alpha(\hat{\Theta}) - \langle \hat{A}, \Theta \rangle, \hat{\Theta})] = \mathbb{E}[\phi_\beta(\langle \hat{A}, \hat{\Theta} \rangle) - \phi_\beta(\langle \hat{A}, \Theta \rangle)] = \mathbb{E}[\phi_\beta(\langle A^\star, \Theta \rangle) - \phi_\beta(\langle \hat{A}, \Theta \rangle)],$$

where we used the fact that the pair $(A^\star, \Theta)$ and $(\hat{A}, \hat{\Theta})$ are identically distributed.

Similarly to the proof in Section 5.2, we construct a function $\varphi_\beta(x, \theta)$ as the tightest upper bound on $\psi_\beta(x, \theta)$ that satisfies the requirements of Lemma 4.

**Definition 3 (Extended logistic surrogate)** *We construct the* extended logistic surrogate *function* $\varphi_\beta(x, \theta)$ *as the tightest upper bound on $\psi_\beta(x, \theta)$ such that $\varphi_\beta(x, \theta)/x$ is non-decreasing over $x \geq 0$ for all $\theta \in \mathcal{O}$.*

*Namely, let $\delta_\beta(\theta) = \arg\max_{x \in [0, 1+\alpha(\theta)]} \frac{\psi_\beta(x, \theta)}{x}$, we define the function $\varphi_\beta(x, \theta)$ as*

$$\varphi_\beta(x, \theta) = \begin{cases} \psi_\beta(x, \theta) & x \in [0, \delta_\beta(\theta)] \\ \psi_\beta(\delta_\beta(\theta), \theta) + (x - \delta_\beta(\theta)) \cdot \psi_\beta(\delta_\beta(\theta), \theta)/\delta_\beta(\theta) & x \in {]}\delta_\beta(\theta), 1 + \alpha(\theta)] \end{cases}.$$

We are now equipped to extend Lemma 5 to general action and parameter spaces.

**Lemma 10** *Let the extended logistic surrogate be defined as in Definition 3. Then, it holds that*

$$\mathbb{E}[\mathrm{Bern}(\phi_\beta(\langle \hat{A}, \Theta \rangle)) - \mathrm{Bern}(\phi_\beta(\langle \hat{A}, \Theta \rangle))]^2 \leq d \cdot \mathbb{E}\left[\mathbb{V}\left[\varphi_\beta(\alpha(\hat{\Theta}) - \langle \hat{A}, \Theta \rangle, \hat{\Theta}) \mid \hat{\Theta}\right]\right].$$

**Proof 10** *The proof follows closely the technique used to prove Lemma 5. We note that conditioned on $\hat{\Theta} = \theta$, the extended logistic surrogate is a mapping from $[0, 1 + \delta_\beta(\theta)]$ to $[0, 1]$, that $\varphi(0, \theta) = \phi_\beta(\alpha(\theta)) - \phi_\beta(\alpha(\theta)) = 0$ and fulfills the assumptions of Lemma 4.*

Noting that $\mathbb{V}\left[\phi_\beta(\langle \hat{A}, \Theta \rangle) \mid \hat{\Theta}\right] = \mathbb{V}\left[\psi_\beta(\alpha(\hat{\Theta}) - \langle \hat{A}, \Theta \rangle, \hat{\Theta}) \mid \hat{\Theta}\right]$ and using Lemma 2, we have that the information ration $\Gamma$ can be bounded by

$$\Gamma \leq d/2 \cdot \frac{\mathbb{E}\left[\mathbb{V}\left[\varphi_\beta\left(\alpha(\hat{\Theta}) - \langle \hat{A}, \Theta \rangle, \hat{\Theta}\right) \mid \hat{\Theta}\right]\right]}{\mathbb{E}\left[\mathbb{V}\left[\psi_\beta\left(\alpha(\hat{\Theta}) - \langle \hat{A}, \Theta \rangle, \hat{\Theta}\right) \mid \hat{\Theta}\right]\right]}.$$

Similarly to the analysis for the $\mathcal{O} \subseteq \mathcal{A}$, we can derive an upper bound by studying the case $\beta \to \infty$ after applying the same preliminary transformations, $f$, $g$, and $h$ as used in Lemma 9 on the functions $\varphi_\beta(x, \theta)$ and $\psi_\beta(x, \theta)$. To express the resulting functions $h(g(f(\psi_\infty(x))))$ and $h(f(\varphi_\infty(x)))$, we need to study the extended logistic function $\psi_\beta(x, \theta)$ and the corresponding extended logistic surrogate $\varphi_\beta(x, \theta)$ for $\beta$ tending to infinity.

Starting with $\psi_\beta(x, \theta)$, we recall that $\psi_\beta(x, \theta) = \phi_\beta(\alpha(\theta)) - \phi_\beta(\alpha(\theta) - x)$ can be written as

$$\psi_\beta(x, \theta) = \frac{1}{1 + \exp(-\beta\alpha(\theta))} - \frac{1}{1 + \exp(-\beta(\alpha(\theta) - x))}.$$

Again, we can distinguish between three cases for $(\alpha(\theta) - x)$: negative, zero, or positive. For values of $x \in {]}\alpha(\theta), 1 + \alpha(\theta)]$, we have that $(\alpha(\theta) - x) < 0$ and that $\lim_{\beta \to \infty} \psi_\beta(x, \theta) = 1$, if $x = \alpha(\theta)$, we have that $\lim_{\beta \to \infty} \psi_\beta(x, \theta) = 1/2$ and for values of $x \in [0, \alpha(\theta)[$, we have that $(\alpha(\theta) - x) > 0$ and that $\lim_{\beta \to \infty} \psi_\beta(x, \theta) = 0$. We can then write

$$\psi_\infty(x, \theta) = \begin{cases} 0 & x \in [0, \alpha(\theta)[ \\ 1/2 & x = \alpha(\theta) \\ 1 & x \in {]}\alpha(\theta), 1 + \alpha(\theta)] \end{cases}.$$

We continue and construct the corresponding $\varphi_\infty(x), \theta$. By definition, we have that $\alpha(\theta) \le 1$ and we note that $\frac{\psi_\infty(x,\theta)}{x}$ is maximized when taking the limit to $x = \alpha(\theta)^+$ from the right: $\lim_{x \to \alpha(\theta)^+} \frac{\psi_\infty(x)}{x} = \frac{1}{\alpha(\theta)}$. It comes that $\varphi_\infty(x)$ can be written as

$$\varphi_\infty(x) = \begin{cases} 0 & x \in [0, \alpha(\theta)[ \\ 1/2 & x = \alpha(\theta) \\ 1 + \frac{x - \alpha(\theta)}{\alpha(\theta)} & x \in ]\alpha(\theta), 1 + \alpha(\theta)] \end{cases}.$$

We can now construct the functions $\overline{\psi}(x, \theta) := h(g(f(\psi_\infty(x, \theta))))$ and $\overline{\varphi}(x, \theta) := h(f(\varphi_\infty(x, \theta)))$ for $f$, $g$, and $h$ defined as in Lemma 9. We note that the resulting functions can be written as

$$\overline{\psi}(x, \theta) = \begin{cases} 0 & x \in [0, \alpha(\theta)] \\ 1 & x \in ]\alpha(\theta), 1 + \alpha(\theta)] \end{cases}, \tag{5}$$

and

$$\overline{\varphi}(x, \theta) = \begin{cases} 0 & x \in [0, \alpha(\theta)] \\ 1 + \frac{2(x - \alpha(\theta))}{\alpha(\theta)} & x \in ]\alpha(\theta), 1 + \alpha(\theta)] \end{cases}, \tag{6}$$

a similar form as the functions $\overline{\psi}$ and $\overline{\varphi}$ be defined respectively in (2) and (3).

**Lemma 11** Let $\overline{\psi}(x, \theta)$ and $\overline{\varphi}(x, \theta)$ be defined respectively in (5) and (6). Then, it holds that

$$\frac{\mathbb{E}\left[ \mathbb{V}\left[ \overline{\varphi}\left( \alpha(\hat{\Theta}) - \langle \hat{A}, \Theta \rangle, \hat{\Theta} \right) \mid \hat{\Theta} \right] \right]}{\mathbb{E}\left[ \mathbb{V}\left[ \overline{\psi}\left( \alpha(\hat{\Theta}) - \langle \hat{A}, \Theta \rangle, \hat{\Theta} \right) \mid \hat{\Theta} \right] \right]} \le \frac{9}{\alpha}.$$

**Proof 11** *The proof follows the one for Lemma 6. Starting with $\mathbb{E}\left[ \mathbb{V}\left[ \overline{\psi}\left( \alpha(\hat{\Theta}) - \langle \hat{A}, \Theta \rangle, \hat{\Theta} \right) \mid \hat{\Theta} \right] \right]$, we note that $\overline{\psi}\left( \alpha(\hat{\Theta}) - \langle \hat{A}, \Theta \rangle, \hat{\Theta} \right)$ is equal to 1 if $\langle \hat{A}, \Theta \rangle < 0$ and is equal to 0 otherwise. It can then be written as $\mathbb{E}[Q(\hat{A})(1 - Q(\hat{A}))]$ using the notations from the proof of Lemma 6. Similarly for $\mathbb{E}\left[ \mathbb{V}\left[ \overline{\varphi}\left( \alpha(\hat{\Theta}) - \langle \hat{A}, \Theta \rangle, \hat{\Theta} \right) \mid \hat{\Theta} \right] \right]$, we distinguish two cases: either $\langle \hat{A}, \Theta \rangle \ge 0$ and the function is equal to 0 or $\langle \hat{A}, \Theta \rangle < 0$ and the function is equal to $1 - 2\frac{\langle \hat{A}, \Theta \rangle}{\langle \hat{A}, \hat{\Theta} \rangle}$. Using a similar decomposition as in we can write the proof of Lemma 6, we can write $\mathbb{E}\left[ \mathbb{V}\left[ \overline{\varphi}\left( \alpha(\hat{\Theta}) - \langle \hat{A}, \Theta \rangle, \hat{\Theta} \right) \mid \hat{\Theta} \right] \right]$ as*

$$\mathbb{E}\left[ I(\langle \hat{A}, \Theta \rangle) \left( 1 - 2\frac{\langle \hat{A}, \Theta \rangle}{\langle \hat{A}, \hat{\Theta} \rangle} \right)^2 \right] - \mathbb{E}\left[ \mathbb{E}\left[ I(\langle \hat{A}, \Theta \rangle) \left( 1 - 2\frac{\langle \hat{A}, \Theta \rangle}{\langle \hat{A}, \hat{\Theta} \rangle} \right) \right]^2 \right],$$

*where in the second term, the outer expectation is on $\hat{A}, \hat{\Theta}$, and the inner expectation is on $\Theta$. Then taking the supremum over the possible values of $1 - 2\frac{\langle \hat{A}, \Theta \rangle}{\langle \hat{A}, \hat{\Theta} \rangle}$ which ranges from $[1 - 2/\alpha, 1 + 2/\alpha]$ we get:*

$$\mathbb{E}\left[ \mathbb{V}\left[ \overline{\varphi}\left( \alpha(\hat{\Theta}) - \langle \hat{A}, \Theta \rangle, \hat{\Theta} \right) \mid \hat{\Theta} \right] \right] \le \sup_{\zeta \in [1 - 2/\alpha, 1 + 2/\alpha]} \mathbb{E}\left[ I(\langle \hat{A}, \Theta \rangle) \zeta^2 \right] - \mathbb{E}\left[ \mathbb{E}\left[ I(\langle \hat{A}, \Theta \rangle) \zeta \right]^2 \right]$$

$$= (1 + 2/\alpha)^2 \cdot \mathbb{E}[Q(\hat{A})(1 - Q(\hat{A}))].$$

*Finally, as $\alpha \in [0, 1]$, we can upper bound $(1 + 2/\alpha)^2$ by $9\alpha^{-2}$ and we conclude the proof.*

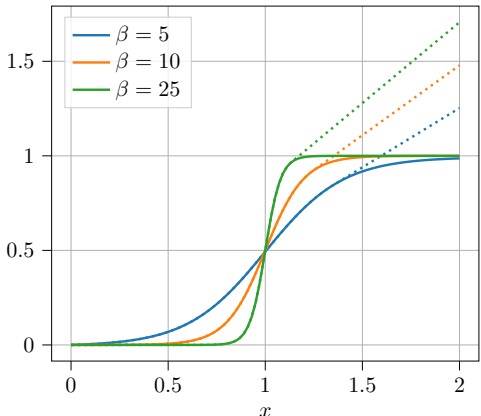

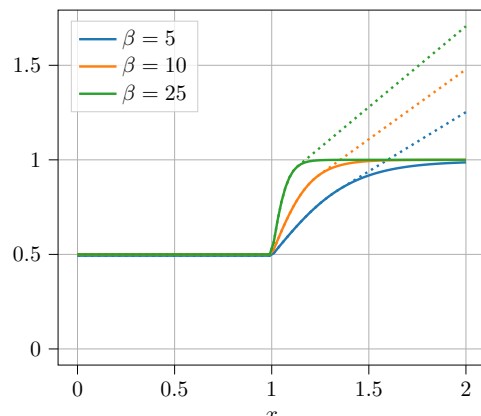

Figure 3: Illustration of the function $\psi_\beta$ (in solid line) and the function $\varphi_\beta$ (in dotted line) for different values of $\beta$.

Figure 4: Illustration of the function $f(\psi_\beta)$ (in solid line) and the function $f(\varphi_\beta)$ (in dotted line) for different values of $\beta$.

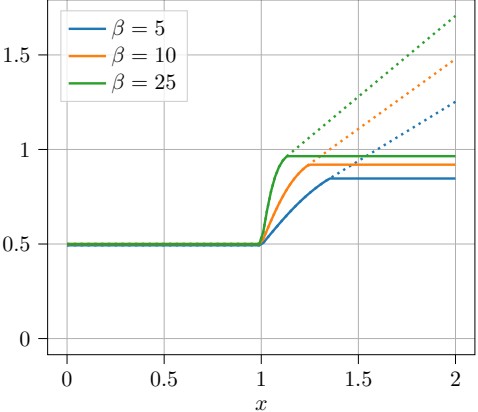

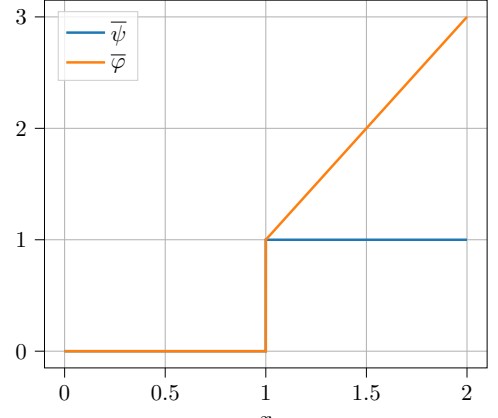

Figure 5: Illustration of the function $g(f(\psi_\beta))$ (in solid line) and the function $f(\varphi_\beta)$ (in dotted line) for different values of $\beta$.

Figure 6: Illustration of the function $\overline{\psi}$ (in blue) and the function $\overline{\varphi}$ (in orange).

## D   NUMERICAL SIMULATIONS

To illustrate the improvement of our regret analysis compared to previous work, we performed numerical experiments on a synthetic problem. We considered a logistic bandit problem in dimension $d = 10$, with time horizon $T = 200$, and with parameter $\beta$ ranging between $[0.25, 10]$. For both action space and parameter space, we used the entire $d$-dimensional unit sphere, $\mathcal{A} = \mathcal{O} = \mathbf{S}_d(0, 1)$ and assumed a uniform prior distribution for the parameter $\Theta$. We computed the expected regret of the Thompson Sampling algorithm using a Markov Chain Monte Carlo (MCMC) method (see Remark 2) and compared it to three Bayesian regret bounds that hold for continuous spaces: (Russo & Van Roy, 2014b, Proposition 4), (Dong & Van Roy, 2018, Theorem 3), and our Corollary 1.

The results are presented in Figure 7. The left subfigure shows the evolution of the expected regret and the regret bounds for the different time steps $t \in \{1, \ldots, 200\}$ and for two different values of $\beta = \{2, 4\}$. For both values of $\beta$, our bound is tighter throughout the entire time horizon and is less sensitive to increasing $\beta$ compared to (Russo & Van Roy, 2014b, Proposition 4) and (Dong & Van Roy, 2018, Theorem 3). This behavior is further illustrated in the right subfigure, were the different regret bounds at $t = 200$ are compared for values of $\beta$ ranging between $[0.25, 10]$. Our bound remains competitive across all values of $\beta$ and is tighter than (Dong & Van Roy, 2018, Theorem 3) for values of $\beta \geq 2$. Importantly, we observe that both (Russo & Van Roy, 2014b, Proposition 4) and (Dong & Van Roy, 2018, Theorem 3) increase exponentially with $\beta$ while our bound increases only logarithmically. We note that the actual expected regret decreases for larger $\beta$. This was anticipated since, for large values of $\beta$, the distinction between near-optimal and suboptimal actions becomes more pronounced, facilitating the identification of near-optimal actions.

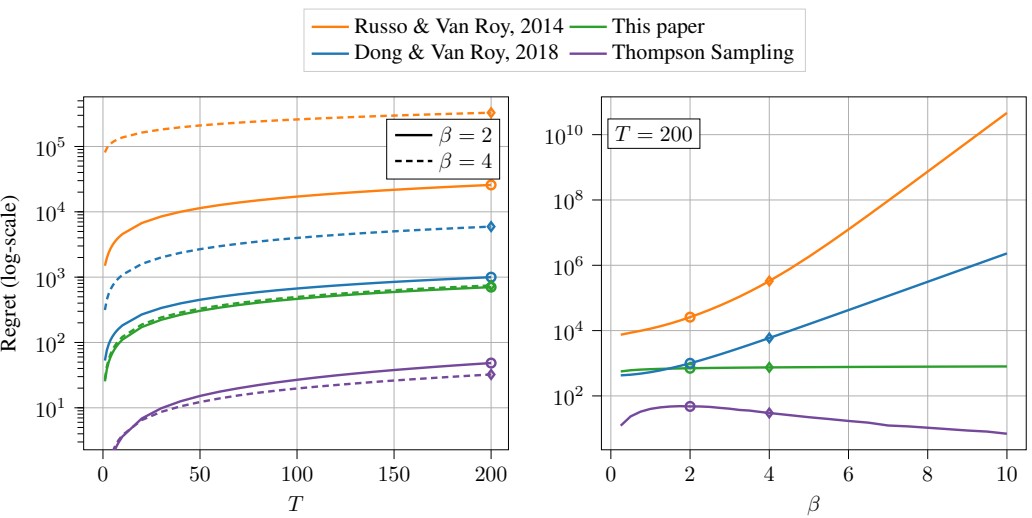

Figure 7: Comparison of Bayesian regret bounds for the logistic bandit setting ($d = 10$, $T = 200$, $\beta \in [0.25, 10]$, $\mathcal{A} = \mathcal{O} = \mathbf{S}_d(0, 1)$ and a uniform prior on $\Theta$). The left subfigure illustrates the evolution of the bounds and the expected regret with respect to the time for two different values of $\beta \in \{2, 4\}$. The right subfigures compares the behavior of the bounds and the expected regret at time $T = 200$ for values of $\beta$ ranging in $[0.25, 10]$. For computing the Thompson Sampling expected regret, we used the MCMC method with 200 particles and averaged over 500 draws.

**Remark 2** *For the logistic bandit setting, we can derive analytically an update rule for the posterior* $\mathbb{P}_t(\Theta)$ *given a new reward* $R_t$ *and the corresponding action* $A_t$, *using Bayes' rule:*

$$\frac{\mathbb{P}_t(\Theta | R_t, A_t)}{\mathbb{P}_t(\Theta)} = \frac{\mathbb{P}_t(R_t | \Theta, A_t)}{\mathbb{P}_t(R_t | A_t)} \propto \mathbb{P}_t(R_t | \Theta, A_t) = \begin{cases} \phi_\beta(\langle \Theta, A_t \rangle) & \text{if } R_t = 1 \\ 1 - \phi_\beta(\langle \Theta, A_t \rangle) & \text{if } R_t = 0 \end{cases},$$

*where the first equality follows from* $\mathbb{P}_t(\Theta) = \mathbb{P}_t(A_t)$ *for Thompson Sampling. For finite parameter spaces, it suffices to update the probability associated with each* $\theta \in \mathcal{O}$ *proportional to* $\mathbb{P}_t(R_t | \Theta, A_t)$ *and normalize afterwards. For continuous spaces, numerical methods such as Markov Chain Monte Carlo (MCMC) or Gibbs Sampling can be used to approximate the distribution of the posterior.*

# E REGARDING THE GAPS IN PREVIOUS LITERATURE

In Section 1, we mention that the main results of Dong et al. (2019) are incomplete because of two shortcomings. The first one concerns a gap in their analysis of the Thompson Sampling information ratio for values of $\beta > 2$. The second one regards a mistake in their regret analysis, which combines incompatible results. We elaborate on both arguments below.

**Regarding the first shortcoming**    We identified an issue in the proof of (Dong et al., 2019, Theorem 5) at the end of the proof on page 20, where the inequality $\chi > \xi > 0.1\lambda$ is stated without justification. This inequality plays a crucial role in deriving their bound on the Thompson Sampling information ratio, but the only evidence provided is (Dong et al., 2019, Figure 3), which illustrates the functions $\chi(\lambda, \beta)$ and $\xi(\lambda, \beta)$ for the specific case of $\beta = 2$. While this figure suggests that the inequality holds for $\beta = 2$, it cannot be used to conclude that the inequality holds in general for $\beta \geq 2$. Additionally, we note that the computation of $\chi(\lambda, \beta)$ and $\xi(\lambda, \beta)$ for given values of $\lambda$ and $\beta$ is highly intricate, and despite our best efforts, we were unable to reproduce (Dong et al., 2019, Figure 3).

**Regarding the second shortcoming**    As discussed earlier, the regret analysis in (Dong et al., 2019, Theorems 1 and 5) combines incompatible results. Specifically, the paper uses a uniform bound on the information ratio of the "standard" Thompson Sampling (provided in (Dong et al., 2019, Appendix B, Eq. (18))) together with (Dong & Van Roy, 2018, Theorem 1), which requires a uniform bound on the information ratio of the *one-step compressed Thompson Sampling*. This inconsistency invalidates the regret bounds derived in (Dong et al., 2019, Theorems 1 and 5). We emphasize that the problem is "hidden" in (Dong et al., 2019, Proposition 9), which incorrectly restates (Dong & Van Roy, 2018, Theorem 4). It is important to note that there is no straightforward way to extend the results from (Dong & Van Roy, 2018, Theorem 4) to (Dong et al., 2019, Proposition 9), that is, to remove the need to bound the *one-step compressed Thompson Sampling*, as this compressed Thompson Sampling is the cornerstone of (Dong & Van Roy, 2018, Theorem 4).

While it is possible to use (Dong & Van Roy, 2018, Proposition 1) directly with a uniform bound on the "standard" Thompson Sampling information ratio, this approach is limited because (Dong & Van Roy, 2018, Proposition 1) provides a loose bound. Specifically, this bound depends on the cardinality of the parameter space $\Theta$ through the entropy $H(\theta^\star)$ (or on the cardinality of the action space through $H(A^\star)$ in the original version (Russo & Van Roy, 2015, Proposition 1)). This issue is highlighted in Dong & Van Roy (2018) on page 3 at the end of Section 3, and serves as a motivation for the introduction of the *one-step-compressed Thompson Sampling* regret analysis in the paper. Combining (Dong & Van Roy, 2018, Proposition 1) with our bound on the Thompson Sampling information ratio, Proposition 1, results in a regret bound of the order $O(\sqrt{dT \log(|\mathcal{O}|)})$, which becomes vacuous for infinite or continuous parameter spaces.

We emphasize that the *one-step compressed Thompson Sampling* information ratio is a fundamentally different quantity and is significantly more challenging to analyze due to the intricate construction of the *one-step compressed Thompson Sampling* (c.f. (Dong & Van Roy, 2018, Proof of Proposition 2)). Notably, the techniques used to analyze the information ratio in Dong et al. (2019), even for the case where $\beta \leq 2$, do not apply to the *one-step compressed Thompson Sampling* information ratio. For instance, (Dong et al., 2019, Proof of Proposition 5) requires a one-to-one mapping between parameters and optimal action (see (Dong et al., 2019, Proof of Proposition 15, eq. (22), inequalities (c) and (e))). However, by definition, the equivalent requirement for *one-step compressed Thompson Sampling* (a one-to-one mapping between the statistic $\psi$ and the optimal action) cannot hold, as by construction the $\psi$ is constructed to be less informative than the parameter $\theta$.

# F CONSTRUCTING $\pi_\star$ AS A ONE-TO-ONE MAPPING

If the mapping $\pi_\star(\theta)$ is not one-to-one, it could either mean that a particular parameter is optimal for several actions or that a particular action is optimal for several parameters.

In the first case, for example, if $a_1, a_2 \in \mathcal{A}$ are optimal for the same parameter $\theta_1 \in \mathcal{O}$, this implies that $\mathbb{E}[R(a_1, \theta_1)] = \mathbb{E}[R(a_2, \theta_1)] \geq \mathbb{E}[R(a, \theta_1)]$ for all $a \in \mathcal{A}$ with $a \neq a_1$ and $a \neq a_2$. In this scenario, we can arbitrarily set $\pi_\star(\theta_1) = a_1$ without affecting the regret of Thompson Sampling, as $\mathbb{E}[R(a_1, \theta_1)] = \mathbb{E}[R(a_2, \theta_1)]$.

In the second case, if an action $a_1 \in \mathcal{A}$ is optimal for multiple parameters, say $\theta_1, \theta_2 \in \mathcal{O}$, we can artificially construct an *action label* set $\mathcal{A}'$ such that two labels, $a_1', a_2' \in \mathcal{A}'$, are associated with $a_1$. For all other actions in $\mathcal{A} \setminus \{a_1\}$, there is a corresponding label in $\mathcal{A}'$. We denote the mapping between action labels and their corresponding actions using the function $\rho : \mathcal{A}' \to \mathcal{A}$. We can construct a function $\pi_\star : \mathcal{O} \to \mathcal{A}'$ such that $\pi_\star$ is a one-to-one mapping between the parameters $\mathcal{O}$ and the action labels $\mathcal{A}'$. We define the *optimal action label* as $A^{\star\prime} = \pi_\star(\Theta)$ and the *Thompson Sampling action label* as ${A_t}' = \pi_\star(\Theta_t)$. This artificial construction, illustrated in Figure 8, is intended solely for the purposes of our regret analysis and has no impact on the regret of Thompson Sampling. The instant regret of Thompson Sampling at time $t \in \{1, \dots, T\}$ remains $R(A^\star, \Theta) - R(A_t, \Theta)$, where $A^\star = \rho(A^{\star\prime})$ is still the optimal action for $\Theta$, and $A_t = \rho(A_t')$ is the action selected by Thompson Sampling for $\Theta_t$. In this context, the Thompson Sampling information ratio would be adapted and defined as:

$$\Gamma_t := \frac{\mathbb{E}_t[R(A^\star, \Theta) - R(A_t, \Theta)]^2}{\mathbb{I}_t(A^{\star\prime}; R(A_t, \Theta), A_t')},$$

representing the ratio between the current squared regret and the information gathered about the optimal action label. One can verify that the analysis of the information ratio in Section 5, Appendix B, and Appendix C, proceeds the same with this adapted definition and leads to the same upper bound.

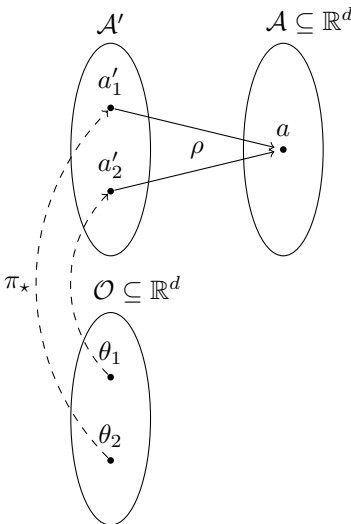

Figure 8: Illustration of the artificial construction of the action label set $\mathcal{A}'$.

