# OpenReview forum: "An Information-Theoretic Analysis of Thompson Sampling for Logistic Bandits"
_ICLR.cc/2025/Conference — Submitted to ICLR 2025_

### Official Review · Reviewer_qHU8 · 2024-11-02

**Soundness:** 3
**Presentation:** 3
**Contribution:** 3
**Rating:** 6
**Confidence:** 4

**Summary:**

The authors derive regret bounds for Thompson sampling applied to a logistic bandit setting. In particular, the authors use the information-ratio technique (Russo & Van Roy) to first upper bound the information ratio, and use that to obtain the regret bound. Crucially, the regret bound is only logarithmically dependent on the "logistic function parameter". Furthermore, though the general regret bound the authors propose has a 1/\alpha term, \alpha being the minmax alignment constant, and this is impossible to remove (as observed in Dong & Van Roy), the authors look at special cases where the dependency can indeed be removed, leading to sub-linear regret bound \tilde{O}(d\sqrt{T}).

**Strengths:**

The main strength of the paper is that it resolves previous gaps in the theory for Thompson sampling applied to logistic bandits. Specifically, this is the first work that has a log dependency on the logistic function parameter, when the number of actions can be large. The authors also show that in the case when the action spaces encompasses the parameter space, the expected regret of Thompson sampling does not depend on the min-max alignment constant, which was previously known to be an obstacle in obtaining good regret bounds.

**Weaknesses:**

I don't have any particular weaknesses in mind, except the compatibility of the paper to the conference. It would also be nice for the authors to talk about some practical implications.

**Questions:**

Can the results be extended to information directed sampling?

---

> ### Author Response · Authors · 2024-11-19
>
> We thank the reviewers for their insightful feedback and are grateful that they found our work resolves previous gaps in the theory for Thompson sampling applied to logistic bandits. We address the reviewer's comments below
> ***
> ___
> **1. Regarding practical applications**
>
> In the Introduction section (lines 33–35), we mention that logistic bandits are used to model personalized advertisement systems. We have now added that logistic bandits are also used for click-through rate prediction in web advertising and for estimating the probability that an email is spam (McMahan, et al., 2012).
> ***
> ___
> **2. Regarding the extension to information directed sampling**
>
> Thank you for this question. Yes, all the results derived in our paper apply directly to the Information Directed Sampling (IDS) algorithm. By definition, at each time step $t \in \{1, \ldots, T\}$, IDS selects the action that minimizes the information ratio (Russo \& Van Roy, 2017). This ensures that the IDS information ratio is less than or equal to the information ratio of Thompson Sampling, allowing us to apply our regret bounds directly to the IDS algorithm.
>
> As discussed in the Conclusion (lines 506–510), an exciting research direction is to extend our analysis to a frequentist version of IDS: the Optimistic Information Directed Sampling (OIDS) algorithm introduced by Neu et al. (2024). We believe this approach could enable the derivation of new frequentist bounds for logistic bandits that do not scale exponentially with the parameter $\beta$, thereby improving on the state-of-the-art bounds in the frequentist setting.
> ***
> ___
> **3. Numerical experiments**
>
> We would like to mention that we have added a section in the Appendix where we performed numerical experiments on a synthetic data problem to illustrate the improvement of our regret analysis compared to previous work. Specifically, we considered a logistic bandit in dimension $d=10$ with continuous action and parameter spaces. We computed the expected regret of the Thompson Sampling algorithm and compared it to three Bayesian regret bounds for logistic bandits: Proposition 4 (Russo \& Van Roy, 2014), Theorem 3 (Dong \& Van Roy, 2018), and our Corollary 1. Our results demonstrate that our bound is tighter throughout the entire time horizon for values of $\beta$ greater than two and increases only logarithmically with $\beta$, whereas Proposition 4 (Russo \& Van Roy, 2014) and Theorem 3 (Dong \& Van Roy, 2018) grow exponentially.
> ***
> ___
> We sincerely appreciate your detailed feedback, which has helped us improve our work. If our responses have adequately addressed your concerns, we hope you consider re-evaluating your score. Should any issues remain, we would be happy to provide further details.

---

### Official Review · Reviewer_SdHQ · 2024-11-04

**Soundness:** 3
**Presentation:** 3
**Contribution:** 3
**Rating:** 6
**Confidence:** 4

**Summary:**

The paper analyses the Thompson Sampling (TS) algorithm applied to logistic bandit problems, where an agent selects actions based on binary rewards determined by a logistic function. The author defines the minimax alignment constant $\alpha:= \min_{\theta \in \mathcal{O}} \max_{a\in \mathcal{A}} <a, \theta>$ to be included in the regret upper bound (also appears in Dong et al., 2019 as the \textit{ worst-case optimal log odds}) where $\mathcal{O}$ is the parameter space and $\mathcal{A}$ is the action space. The final regret bound is $O(d/\alpha \cdot T^{1/2})$ ignoring the logarithmic factor, where $d$ is the dimension of the parameter space and $T$ is the time horizon. The author improves the regret bound shown in Dong et al., which is believed to be the most relevant proceeding work, by dropping the fragility dimension $\eta$. Since $\eta$ could go exponentially with dimension $d$, dropping the dependency on $\eta$ can tighten the regret upper bound.
The regret bound present in this work is also believed to be the first regret bound for logistic bandits that achieve $\log(\beta)$ dependency.

**Strengths:**

The paper is well-written with clear explanations and well-structured arguments that enhance the accessibility of complex ideas. The upper bound of regret is tighter than the most advanced result in the literature. The proof stream is clear to the reader.

**Weaknesses:**

In comparing results from multiple works (table 1), Dong et al. have not been included, while it should be an important preceding since some major concepts are borrowed from there, such as the minimax alignment constant. Essentially, the work should emphasize the improvement of Dong et al.

Also, the details of the Thompson Sampling algorithm could be elaborated more, such as how the posterior sampling distribution has been established and how the parameters have been updated once the agent collects a new reward.

**Questions:**

Please refer to the weaknesses.

**Details Of Ethics Concerns:**

Non ethics concerns.

---

> ### Author Response · Authors · 2024-11-19
>
> We thank the reviewer for their insightful feedback and are grateful that they found our explanation clear and with well-structured arguments. We appreciate they found our results to be tighter than the most advanced results in the literature. We address the reviewer's comments below.
> ___
>
>
> **1. Regarding the results of (Dong et al., 2019) not figuring in Table 1**
>
>
>  Since the regret analysis of (Dong et al., 2019) Theorem 1 is incorrect, we believe it would be misleading to include it in the comparison in Table 1. As highlighted in the Introduction (lines 78-83), the regret analysis in (Dong et al., 2019) (Theorems 1 and 5) combines incompatible results. Specifically, they apply a uniform bound on the information ratio of the "standard" Thompson Sampling (provided in Appendix B, equation 18) together with (Dong \& Van Roy, 2018) Theorem 1,  which requires a uniform bound on the information ratio of the "one-step compressed Thompson Sampling". This mistake invalidates the regret bounds they derive in Theorems 1 and 5.
>
> We emphasize that the problem is "hidden" in (Dong et al., 2019) Proposition 9, which incorrectly restates (Dong \& Van Roy, 2018) Theorem 4. It is important to note that the "one-step compressed Thompson Sampling" information ratio is a fundamentally different quantity and is, because of the intricate construction of the compressed Thompson Sampling (c.f. (Dong \& Van Roy, 2018) Appendix A, Proof of Proposition 2), significantly more challenging to analyze. Notably, the techniques used in (Dong et al., 2019) to bound the standard information ratio do not apply to the one-step compressed Thompson Sampling information ratio. For instance, the Proof of Proposition 5 in (Dong et al., 2019) requires a one-to-one mapping between parameters and optimal action (see Proof of Proposition 15, eq. (22), inequalities (c) and (e) in (Dong et al., 2019)). However, by definition, the equivalent requirement for *one-step compressed Thompson Sampling* (a one-to-one mapping between the statistic $\psi$ and the optimal action) cannot hold, as by construction the $\psi$ is constructed to be less informative than the parameter $\theta$.
>
> We have added the above explanation a new appendix section, Appendix E, which can be found in the revised PDF.
> ___
>
>
> **2. Regarding the details of TS posterior update for logistic bandits**
>
> For the logistic bandit setting, we can derive analytically an update rule for the posterior $\mathbb{P}_t(\Theta)$ given a new reward $R_t$ and the corresponding action $A_t$, using Bayes' rule:
>         $\frac{\mathbb{P}_t(\Theta|R_t,A_t)}{\mathbb{P}_t(\Theta)} = \frac{\mathbb{P}_t(R_t|\Theta,A_t)}{\mathbb{P}_t(R_t|A_t)} \propto \mathbb{P}_t(R_t|\Theta,A_t) $ where $\mathbb{P}_t(R_t |\Theta, A_t)$
>
> is equal to
> $ \phi_{\beta}(\langle \Theta, A_t \rangle)$ if $R_t=1$
> and equal to $1-\phi_\beta(\langle \Theta, A_t\rangle)$ if $R_t=0$.
>
> For finite parameter spaces, it suffices to update the probability associated with each $\theta\in\mathcal{O}$ proportional to $\mathbb{P}_t(R_t|\Theta,A_t)$ and normalize afterward. For continuous spaces, numerical methods such as Markov Chain Monte Carlo (MCMC) or the Gibbs Sampling algorithm can be used to approximate the distribution of the posterior. We have added this remark in the appendix section about numerical simulation.
>
> ___
>
>
> **3. Numerical experiments**
>
> We would like to mention that we have added a section in the Appendix where we performed numerical experiments on a synthetic data problem to illustrate the improvement of our regret analysis compared to previous work. Specifically, we considered a logistic bandit in dimension $d=10$ with continuous action and parameter spaces. We computed the expected regret of the Thompson Sampling algorithm and compared it to three Bayesian regret bounds for logistic bandits: Proposition 4 (Russo \& Van Roy, 2014), Theorem 3 (Dong \& Van Roy, 2018), and our Corollary 1. Our results demonstrate that our bound is tighter throughout the entire time horizon for values of $\beta$ greater than two and increases only logarithmically with $\beta$, whereas Proposition 4 (Russo \& Van Roy, 2014) and Theorem 3 (Dong \& Van Roy, 2018) grow exponentially.
>
> ___
>
> We sincerely appreciate your detailed feedback, which has helped us improve our work. If our responses have adequately addressed your concerns, we hope you consider re-evaluating your score. Should any issues remain, we would be happy to provide further details.

---

> > ### Comment · Reviewer_SdHQ · 2024-12-02
> >
> > Thanks for your detailed feedback. I like the newly added section which explains the justification made by the author more clearly. I will keep my score and support to this paper.

---

> > > ### Author Response · Authors · 2024-12-04
> > >
> > > Thank you for your response. We are grateful that we could address all your questions and that you found the newly added appendix sections helpful. We sincerely appreciate your support for our paper.

---

### Official Review · Reviewer_MVqR · 2024-11-04

**Soundness:** 3
**Presentation:** 3
**Contribution:** 3
**Rating:** 6
**Confidence:** 3

**Summary:**

The paper studies the performance of Thompson sampling for logistic bandit problems and improves upon previous results.

**Strengths:**

Using the information theoretic framework of (Russo & Van Roy, 2015), this paper improves upon the previous regret bounds for Thompson sampling in logistic bandit problems. In particular, it removes dependence on the "fragility dimension" which is known to grow exponentially in some cases.

**Weaknesses:**

1. In Lines 77 to 83, it is claimed that the main result of (Dong et al, 2019) is incorrect. This is a strong claim that requires an equally strong proof.
First, it is claimed that they do not provide a rigorous proof for generalizing their bound to larger values of $\beta$.
However, there is no discussion about the specifics of this proof or why it might not be rigorous.
Second, it is claimed that their regret analysis relies on the rate-distortion bound in (Dong & Van Roy, 2018) which specifically requires a bound on the "one-step compressed Thompson sampling information ratio".
However, the result of (Dong & Van Roy, 2018) that is used in (Dong et al, 2019) is stated in the form of Proposition 9 in (Dong et al, 2019) and does not require such bounds.
It is true that this result, as it is written does not appear in (Dong & Van Roy, 2018).
Proposition 9 in (Dong et al, 2019) is based on Theorem 4 in (Dong & Van Roy, 2018) which uses the notion of one-step compressed information ratio.
However, when the (normal notion of) information ratio is bounded, then the result seems to be simpler to prove.
In fact, looking at the proof of Theorem 4 in (Dong & Van Roy, 2018), it seems that the the notion of one-step compressed information ratio only appears through Conjecture 1 and if we instead have a uniform bound on information ratio, then we could directly use Proposition 1 instead of Theorem 1 and obtain the result stated in (Dong et al, 2019).
If my understanding is correct, then the mistake in (Dong et al, 2019) is that they didn't include an argument in the spirit of what I described, but their result holds.

2. Proposition 11 in (Dong et al, 2019) states that no regret bound of the form $f(\alpha)p(d)T^{1 - \epsilon}$ is possible.
This does not imply that removing dependence on alpha is not possible.
A claim that is mentioned both in the paragraph from lines 292 to 298 and in the conclusion section.
Instead, what it implies is that removing dependence on both beta and eta *at the same time* (without introducing other problem-dependent terms) is not possible.
The regret bound of (Dong et al, 2019), specifically in their Theorem 5, removes dependence on beta, but depends on eta. (which in some cases may grow exponentially)
Your regret bound, in Theorem 2, removes dependence on eta, but depends on beta.

3. In the case A = O, Corollary 2 seems to be a weaker version of Theorem 1 in (Dong et al, 2019).
A sentence should be added next to the Corollary to discuss how it compares to that theorem.

**Questions:**

Why can we assume that the mapping $\pi_*$ is one-to-one?
This is mentioned in the footnote of page 3, but this argument seems to be about settings where we could duplicate actions.
Is this trivial that we could do that in the logistic bandit setting?

---

> ### Author Response · Authors · 2024-11-19
>
> We thank the reviewers for their insightful feedback and are grateful that they found that our paper improves upon the previous regret bounds for Thompson sampling in logistic bandit problems. We address the reviewer's comments below.
> * * *
> ____
> **1.a. Regarding (Dong et al., 2019) non-rigorous analysis of the TS information ratio**
>
>  We agree that our claim (on line 77) that "(Dong et al., 2019) did not provide a proof on TS information ratio for $\beta\geq 2$" requires a more detailed explanation. To be precise, we noticed an issue in the proof of (Dong et al., 2019) Thm 5, at the end of the proof on page 20, where the inequality $\chi > \xi > 0.1 \lambda$ is provided without justification. This inequality plays a crucial role in deriving their bound on the TS information ratio, but the only evidence that it might hold is (Dong et al., 2019) Fig 3, which illustrates the functions $\chi(\lambda,\beta)$ and $\xi(\lambda,\beta)$ for the case $\beta=2$. This figure is an example that the inequality seems to hold for $\beta=2$, but it cannot be used to infer that it generally holds for $\beta\geq 2$. We can add the above explanation to the Appendix.
> * * *
> ____
> **1.b. Regarding using (Dong \& Van Roy, 2018) Prop 1 together with uniform bounds on the TS information ratio**
>
> Thank you for your detailed analysis. While it is possible to use (Dong \& Van Roy, 2018) Prop 1 directly with a uniform bound on the normal TS information ratio, this approach is limited because Prop 1 is loose. Indeed, the bound in Prop 1 depends on the cardinality of the parameter space $\Theta$ through the entropy $\mathsf{H}(\theta^\star)$ (or on the cardinality of the action space through $\mathsf{H}(A^\star)$ in the original version of Prop 1 in (Russo \& Van Roy, 2014)). This issue is explained in (Dong \& Van Roy, 2018) on page 3 at the end of Section 3, and motivates the introduction of the "one-step-compressed TS" regret analysis in their paper. Combining (Dong \& Van Roy, 2018) Prop 1 with our bound on the TS information ratio (our Prop 1) results in a regret bound of the order $O(\sqrt{d T \log(|\mathcal{O}|)})$, which becomes vacuous for infinite or continuous parameter spaces.
> * * *
> ____
>
> **2. Regarding the dependency on $\alpha$ and the fragility dimension**
>
> You are absolutely right, and we appreciate your careful observation. We missed this subtlety in our original manuscript and have now corrected lines 292 to 298 as well as the conclusion accordingly.
>
> * * *
> ____
>
> **3. Regarding comparing our Corollary 1 with (Dong et al., 2019) Thm 1**
>
> Since the regret analysis of (Dong et al., 2019) Thm 1 is incorrect, it would be misleading to compare it to our Corollary 1.
>  As highlighted in the Introduction (lines 78-83), the regret analysis in (Dong et al., 2019) (Thm 1 and 5)  combines incompatible results. Specifically, they apply a uniform bound on the information ratio of the "standard" TS (provided in Appendix B, equation 18), together with (Dong \& Van Roy, 2018) Thm 1,  which requires a uniform bound on the information ratio of the "one-step compressed TS". This mistake invalidates the regret bounds they derive in Thm 1 and 5. We want to emphasize that the "one-step compressed TS" information ratio is a fundamentally different quantity and is significantly more challenging to analyze.
> * * *
> ____
> **Question regarding the mapping $\pi^\star$ being one-to-one**
>
> The observation that it is possible to work with an equivalent action space such that the mapping $\pi^\star$ is one-to-one is not specific to the logistic bandits setting. If multiple actions, for example, $a_1 \in \mathcal{A}$ and $a_2 \in \mathcal{A}$, are optimal for the same parameter $\theta \in \mathcal{O}$, meaning that $\mathbb{E}[R(a_1,\theta)] = \mathbb{E}[R(a_2,\theta)] \geq \mathbb{E}[R(a,\theta)]$ for all $a \in \mathcal{A}$, $a \neq a_1$, $a \neq a_2$, we can arbitrarily choose $\pi_\star(\theta) = a_1$ without affecting the regret of Thompson Sampling.
> Similarly, if an action $a \in \mathcal{A}$ is optimal for multiple parameters, such as $\theta_1 \in \mathcal{O}$ and $\theta_2 \in \mathcal{O}$, we can artificially augment the action set $\mathcal{A}$ by adding a duplicate instance of $a$, i.e. $a'=a$, and arbitrarily assign $\pi_\star(\theta_1) = a$ and $\pi_\star(\theta_2) = a'$. This construction is intended solely for analytical purposes.
> * * *
> ____
> **Numerical experiments**
>
> We have added a section in the Appendix where we performed numerical experiments that illustrate the improvement of our regret analysis compared to previous work.
> * * *
> ____
> We sincerely appreciate the detailed feedback, which has helped us improve our work through additional clarifications.If our responses have adequately addressed your concerns, we hope you consider re-evaluating your score.  For any remaining issues, we would be happy to provide further details.

---

> > ### Comment · Reviewer_MVqR · 2024-11-26
> >
> > Thank you for your response.
> >
> > 1- My initial concern was that whether the issue you've raised about (Dong et al, 2019) is because they "left parts of the proof as an exercise to the reader and therefore their proof is incomplete" or "their argument is not complete and the gaps are neither trivial nor easily completable". I'm convinced now that in both cases, the gaps in their proof is not trivial and not easily completable.
> > Given that we are talking about the main result of a published paper, I believe it's essential to include a more detailed discussion, possibly in the appendix. It doesn't have to go through all the details of previous proofs, just a few paragraphs of explanation, basically a short high level explanation, i.e. what you have already written in the paper and your response here, and also a short low level explanation, i.e. an explanation of how the propositions and theorems in those papers fit together and why the proof of Theorem 4 in (Dong & Van Roy, 2018) does not easily translate into a proof for Proposition 9 in (Dong et al., 2019).
> > Part of the reason I wasn't immediately convinced was that there was no low level explanation, so I had to go through the proofs a bit and find out which exact theorems we are talking about, which proposition fails and how.
> >
> > 3- Now that I agree your argument about the previous point, I agree with this point, too. However, i still think it's good to include a sentence or two to mention that result and then mention that their proof is incomplete and refer to the section in the appendix where you go into more details, i.e. the new appendix section I mentioned above.
> >
> > Q- I agree that if there are multiple actions, say $a_1, a_2$ that are optimal for the same parameter $\theta$, we may choose any action we want. My question is about the second part. If an action $a$ is optimal for multiple parameters $\theta_1, \theta_2$ how exactly can we add actions to our action set and still keep the problem a *logistic bandit* problem? If we just add points arbitrarily, I'm not sure the resulting problem will necessarily be a logistic bandit. If the action set $A$ was previously a subset of $\mathbb{R}^d$, would it now be a subset of $\mathbb{R}^d \times \\{0, 1\\}$ where $A \times \\{0\\} \subseteq \mathbb{R}^d \times \\{0\\}$ corresponds to the original actions and $(a, 1) \in \mathbb{R}^d \times \\{1\\}$ corresponds to the new action we just added? How could such a problem be formulated as a new logistic bandit with an action set belonging to $\mathbb{R}^{d'}$ for some $d'$?

---

> ### Author Response · Authors · 2024-11-28
>
> We thank the reviewer for the follow-up questions. We address each question below.
> ***
> ____
> **(1,3) Regarding the gaps in previous literature**
>
> We are pleased that our previous response addressed your concerns. We have now added a new appendix section, Appendix E, which elaborates further on the gaps in (Dong et al., 2019). This section restates the explanation provided in our previous response and includes a detailed, low-level analysis identifying which part of the proof in (Dong et al., 2019) fails and why. The new appendix section can be found in the revised PDF.
> ***
> ____
> **Question regarding the mapping $\pi^\star$ being one-to-one**
>
> Regarding the mapping $\pi_\star$, we agree that the construction of "duplicated" actions requires further explanation. To make the argument rigorous, we distinguish between the action set $\mathcal{A} \subseteq \mathbb{R}^d$ and the artificially constructed set $\mathcal{A}'$, which is an arbitrary set of action labels corresponding to elements in $\mathcal{A}$.
>
>  To be precise, if an action $a_1 \in \mathcal{A}$ is optimal for multiple parameters, say $\theta_1, \theta_2 \in \mathcal{O}$, we can artificially construct an *action label* set $\mathcal{A}'$ such that two labels, $a_1', a_2' \in \mathcal{A}'$, are associated with $a_1$. For all other actions in $\mathcal{A} \setminus \{a_1\}$, there is a corresponding label in $\mathcal{A}'$. We denote the mapping between action labels and their corresponding actions using the function $\rho: \mathcal{A}' \to \mathcal{A}$.
>
> We can construct a function $\pi_\star: \mathcal{O} \to \mathcal{A}'$ such that $\pi_\star$ is a one-to-one mapping between the parameters $\mathcal{O}$ and the action labels $\mathcal{A}'$. We define the *optimal action label* as ${A^\star}' = \pi_\star(\Theta)$ and the *Thompson Sampling action label* as ${A_t}' = \pi_\star(\Theta_t)$.
>
> This artificial construction is intended solely for the purposes of our regret analysis and leaves the logistic bandit problem unchanged. It has no impact on the regret of Thompson Sampling, which remains $R(A^\star, \Theta) - R(A_t, \Theta)$, where $A^\star = \rho({A^\star}')$ is still the optimal action for $\Theta$, and $A_t = \rho(A_t')$ is the action selected by Thompson Sampling for $\Theta_t$.
>
> In this context, the Thompson Sampling information ratio would be adapted and defined as: $$\Gamma_t \coloneqq \frac{\mathbb{E}_t[R(A^\star, \Theta) - R(A_t, \Theta)]^2}{\mathbb{I}_t({A^\star}'; R(A_t, \Theta), A_t')},$$
>  representing the ratio between the current squared regret and the information gathered about the *optimal action label*. One can verify that the analysis of the information ratio in Section 5, Appendix B, and Appendix C, proceeds the same with this adapted definition and leads to the same upper bound.
>
> We have added the above explanation in a new appendix section, Appendix F, and illustrated the artificial construction of the action label set $\mathcal{A}'$ in Figure 8.
> ***
> ____
> We sincerely appreciate your follow-up questions, which have helped us improve our work. If our responses have adequately addressed your concerns, we hope you consider re-evaluating your score. Should any issues remain, we would be happy to provide further clarification.

---

> > ### Comment · Reviewer_MVqR · 2024-12-03
> >
> > Thank you for your detailed response and adding the sections in the appendix. I will increase my score.

---

> > > ### Author Response · Authors · 2024-12-04
> > >
> > > Thank you for your response. We are grateful that we could address all your questions and that you found the new appendix sections helpful. We sincerely appreciate your support for our paper.

---

### Official Review · Reviewer_gokE · 2024-11-07

**Soundness:** 3
**Presentation:** 3
**Contribution:** 2
**Rating:** 5
**Confidence:** 3

**Summary:**

This paper investigates Thompson Sampling (TS) in the context of logistic bandit problems, where rewards are binary and determined by a logistic function. The authors analyze the information ratio to provide a refined regret bound that scales logarithmically with the slope parameter of the logistic function, $\beta$, independent of the number of actions. This approach purportedly improves upon prior bounds that exhibit exponential dependency on $\beta$. Additionally, the paper presents conditions under which dependence on certain alignment constants can be reduced, along with theoretical regret bounds based on this framework.

**Strengths:**

1: The paper offers a refined theoretical analysis for Thompson Sampling in logistic bandits, claiming improved regret bounds that scale logarithmically with $\beta$.

2: The use of information-theoretic concepts, particularly the information ratio, is well-aligned with recent trends in bandit research and has the potential to contribute to a deeper understanding of the regret of logistic bandits.

**Weaknesses:**

1. Lack of technical novelty: while the paper presents a refined analysis, the techniques used are predominantly based on established methods. The information-theoretic approach and analysis of TS have been well-studied in prior work. The paper could benefit from highlighting the unique challenges or technical hurdles in applying these methods to logistic bandits, which are not sufficiently emphasized.

2. Narrow scope: The scope of the study is somewhat limited to logistic bandits without any exploration into a broader class, such as generalized linear models.

3. The paper does not introduce any new algorithms or propose an experimental validation to support its theoretical claims.

4. Bayesian Regret Bounds: The preference for Bayesian regret bounds over frequentist bounds may not appeal to other audiences.

**Questions:**

See weaknesses.

---

> ### Author Response · Authors · 2024-11-19
>
> We thank the reviewers for their insightful feedback and are grateful that they found that our paper offers a refined theoretical analysis for Thompson Sampling in logistic bandits.  We address the reviewer's comments below.
> ***
> ____
> **1. Regarding the technical hurdles and the unique challenges coming with logistic bandits**
>
> As discussed in the Introduction (lines 39-42), a difficulty inherent to the logistic bandit setting is the degree of non-linearity of the logistic function $\phi_\beta(x)$. More specifically, adapting the methods developed for linear bandits to the logistic setting ((Russo \& Van Roy, 2014), (Dong \& Van Roy, 2018)) automatically yields regret bounds that depend on the ratio between the largest and the smallest derivative of the logistic function, $((1+\exp(\beta))^2)/(4\exp(\beta))$, which grows exponentially with the parameter $\beta$. Such a dependence is highly unsatisfactory because, in practice, as $\beta$ increases, it becomes easier to identify the optimal action. Prior to our work, all existing results on logistic bandits presented either an exponential dependence on $\beta$ or a scaling with the cardinality of the action space.
>  The real challenge was removing both the exponential dependency $\beta$ and the scaling on the cardinality of the action space $|\mathcal{A}|$. To do so, we strengthened methods introduced by (Dong et al., 2019) and showed in Section 5 that the TS information ratio could be controlled via a ratio based on the expected variance of the reward probability, conditioned on the sampled action, $\mathbb{E}[\\mathbb{V}[\phi_\beta(\langle\hat{A},\Theta\rangle)|\hat{A}]]$ (lines 442-450, Section 5).
> The key idea in our analysis was to use the limit case $\beta \to \infty$ to derive a general upper bound for all $\beta>0$. This technical novelty is presented in both Section 5.3 and Appendix B.
> ***
> _____
> **2. Regarding the generalization to Generalized Linear Models**
>
> We agree that extending our analysis to the broader class of generalized linear bandits is a promising direction for future work. As explained in the Conclusion (lines 503-506), we believe that this paper lays some important first steps towards improved bounds for the class of Generalized Linear Bandits. The properties of the logistic function that we leveraged in our regret analysis (i.e. $\phi_\beta(x)$ being strictly increasing and having a corresponding surrogate function as defined in Definition 2) could be shared by other classes of link function and be used to derive regret bounds using a similar analysis of the information ratio as we performed in Section 5. This extension is an exciting avenue for future research.
> ***
> ___
> **3. Regarding experimental validation**
>
> We have added a new section in the Appendix, where we performed numerical experiments on a synthetic data problem to illustrate the improvement of our regret analysis compared to previous work. Specifically, we considered a logistic bandit in dimension $d=10$, with continuous action and parameter spaces. We computed the expected regret of the Thompson Sampling algorithm and compared it to three Bayesian regret bounds for logistic bandits: Proposition 4 (Russo \& Van Roy, 2014), Theorem 3 (Dong \& Van Roy, 2018), and our Corollary 1. Our results show that our bound is tighter throughout the entire time horizon for values of $\beta$ greater than two and increases only logarithmically with $\beta$. In contrast, Proposition 4 (Russo \& Van Roy, 2014) and Theorem 3 (Dong \& Van Roy, 2018) grow exponentially.
> ***
> _____
> **4. Regarding Bayesian over frequentist bounds**
>
> As discussed in the Conclusion (lines 506-510), a very exciting research direction is to use the result in this paper to derive frequentist regret bounds. A promising way to do so is to apply our information theoretic analysis to the optimistic information directed sampling algorithm introduced by (Neu et al., 2024). We believe this approach could lead to deriving new frequentist bounds for logistic bandits that would not scale exponentially with the parameter $\beta$, thus improving the state-of-the-art bounds in the frequentist setting.
> ***
> ___
>
> We sincerely appreciate your detailed feedback, which has helped us improve our work. If our responses have adequately addressed your concerns, we hope you consider re-evaluating your score. Should any issues remain, we would be happy to provide further details.

---

> > ### Comment · Reviewer_gokE · 2024-12-03
> >
> > Thank you for your response. I have no further questions. I will maintain my score, as I believe the techniques in this paper are not particularly novel. That said, I am not opposed to its acceptance.

---

> > > ### Author Response · Authors · 2024-12-04
> > >
> > > Thank you for your response. We are grateful that we could address all your questions. We want to reiterate that our paper improves on all prior theoretical analyses for logistic bandits and provides the first bound that explains the performance of any algorithm across all possible logistic functions. We regret that the techniques employed to derive these novel results may have been perceived as a limitation. We appreciate your openness to the paper's acceptance and hope this can be reflected in your final score.

---

### Author Response · Authors · 2024-11-29
**Comment to all reviewers**

We sincerely thank the reviewers for their valuable and constructive feedback. In response to the discussion, we have added three new and insightful appendix sections to the paper:
- *Appendix D* illustrates the improvement of our regret analysis over previous works through numerical experiments on a synthetic logistic bandit problem.  Our bound remains competitive across all values of $\beta$, outperforming (Russo & Van Roy, 2014, Proposition 4) overall and (Dong & Van Roy, 2018, Theorem 3) for $\beta \geq 2$. Importantly, we observe that both (Russo & Van Roy, 2014, Proposition 4) and (Dong & Van Roy, 2018, Theorem 3) increase exponentially with $\beta$, whereas our bound grows only logarithmically. This section also provides details regarding the update of the Thompson Sampling posterior for logistic bandits.
- In *Appendix E*, we elaborate on the gaps in (Dong et al., 2019) and explain how these gaps invalidate their regret analysis. We provide both high- and low-level analyses to clearly identify which part of the proof in (Dong et al., 2019) fails and why.
- Lastly, *Appendix F* rigorously explains the construction of $\pi_\star$ to ensure it is a one-to-one mapping.

We sincerely appreciate the detailed feedback, which has helped us improve our work through additional clarifications. If our responses have adequately addressed your concerns, we hope you will consider re-evaluating your score. Should there be any remaining issues, we would be happy to provide further details.

---

### Meta-Review · Area_Chair_FkBg · 2024-12-21

**Metareview:**

This paper investigates Thompson Sampling (TS) in the context of logistic bandit problems, where rewards are binary and determined by a logistic function. The authors analyze the information ratio to derive a refined regret bound that scales logarithmically with the slope parameter of the logistic function, offering an improvement over previous results from Dong & Van Roy (2018) and Dong et al. (2019).

While the reviewers acknowledged the improvement in regret bounds, they also noted significant limitations in the paper. These include a lack of technical novelty, limited empirical evaluation, and narrow applicability restricted to logistic bandits, without extending to broader bandit classes. Additionally, the paper misses an important reference: Kveton et al. (2020), Randomized Exploration in Generalized Linear Bandits. This work provides strong frequentist regret guarantees for generalized linear bandits using Thompson Sampling, a setting that subsumes the logistic bandit problem studied in this paper.

Given the lack of novel techniques, the limited scope of the work, and the omission of a critical related paper, I recommend rejection in its current form. I encourage the authors to carefully elaborate on how their results compare with the frequentist regret analysis of Thompson Sampling for generalized linear bandits, which is known to be both more general and more challenging than the setting addressed here.

**Additional Comments On Reviewer Discussion:**

The reviewers raised two key concerns: insufficient technical novelty and limited comparative analysis with selected papers. The authors' response fails to adequately address these fundamental issues following the discussion.

---

### Decision · Program_Chairs · 2025-01-22

Reject